# Multimodal deep learning to predict prognosis in adult and pediatric brain tumors

Sandra Steyaert [1,5], Yeping Lina Qiu[1,2,5], Yuanning Zheng[1], Pritam Mukherjee[1], Hannes Vogel[3] & Olivier Gevaert [1,4 ✉]

**Abstract**

**Background** The introduction of deep learning in both imaging and genomics has significantly advanced the analysis of biomedical data. For complex diseases such as cancer, different data modalities may reveal different disease characteristics, and the integration of imaging with genomic data has the potential to unravel additional information than when using these data sources in isolation. Here, we propose a DL framework that combines these two modalities with the aim to predict brain tumor prognosis.

**Methods** Using two separate glioma cohorts of 783 adults and 305 pediatric patients we developed a DL framework that can fuse histopathology images with gene expression profiles. Three strategies for data fusion were implemented and compared: early, late, and joint fusion. Additional validation of the adult glioma models was done on an independent cohort of 97 adult patients.

**Results** Here we show that the developed multimodal data models achieve better prediction results compared to the single data models, but also lead to the identification of more relevant biological pathways. When testing our adult models on a third brain tumor dataset, we show our multimodal framework is able to generalize and performs better on new data from different cohorts. Leveraging the concept of transfer learning, we demonstrate how our pediatric multimodal models can be used to predict prognosis for two more rare (less available samples) pediatric brain tumors.

**Conclusions** Our study illustrates that a multimodal data fusion approach can be successfully implemented and customized to model clinical outcome of adult and pediatric brain tumors.

**Plain Language Summary**

An increasing amount of complex patient data is generated when treating patients with cancer, including histopathology data (where the appearance of a tumor is examined under a microscope) and molecular data (such as analysis of a tumor's genetic material). Computational methods to integrate these data types might help us to predict outcomes in patients with cancer. Here, we propose a deep learning method which involves computer software learning from patterns in the data, to combine histopathology and molecular data to predict outcomes in patients with brain cancers. Using three cohorts of patients, we show that our method combining the different datasets performs better than models using one data type. Methods like ours might help clinicians to better inform patients about their prognosis and make decisions about their care.

[1] Stanford Center for Biomedical Informatics Research (BMIR), Department of Medicine, Stanford University, Stanford, CA, USA. [2] Department of Electrical Engineering, Stanford University, Stanford, CA, USA. [3] Department of Pathology, Stanford University, Stanford, CA, USA. [4] Department of Biomedical Data Science, Stanford University, Stanford, CA, USA. [5] These authors contributed equally: Sandra Steyaert, Yeping Lina Qiu. ✉email: ogevaert@stanford.edu

Glioma is the most prevalent brain tumor type, accounting for almost 80% of all malignant primary brain tumors[1]. Gliomas are classified by their origin and usually graded based on their behavior and/or histopathological features using the World Health Organization (WHO) Classification of Tumors of the Central Nervous System (CNS), the international standard for brain and spinal cord tumor grading[2,3]. In 2016, the WHO revisited these guidelines recommending histopathological diagnosis in combination with molecular markers (e.g. *IDH1/2* mutation status) to classify gliomas and other CNS tumors. For example, gliomas with *IDH* mutations are distinct from *IDH*-wildtype gliomas with different prognosis and these molecular features are now also reflected in the WHO grading[2,4,5]. Of late, DNA-methylation has also been suggested as an additional feature to classify CNS tumors[6,7]. Brain tumors are graded on a scale from 1 to 4, but gliomas are mostly classified as either high-grade glioma (HGG), including grades 3 and 4 gliomas—which are highly malignant with overall bad prognosis—or low-grade glioma (LGG), including grades 1 and 2—which are less aggressive and have better prognosis but may develop into HGG at a later stage. The HGG group mainly consists of Glioblastoma multiforme (GBM), the most aggressive malignant brain tumor, accounting for 60% of all brain tumors in adults[1].

In infants, children or young adolescents, the majority of brain tumors are LGGs. These pediatric LGGs are fundamentally different from adult LGGs in that they rarely develop into malignant HGGs and generally show excellent survival outcome[8]. HGGs are less common in this young population, representing only 8–12% of pediatric CNS tumors, with GBM and anaplastic astrocytoma being the most common ones[9]. Other pediatric brain tumors are medulloblastoma and ependymoma, two malignant subtypes. Malignant brain tumors are one of the highest mortality causes in the pediatric cancer population. For this reason, their study has attracted wide clinical interest, but pediatric brain tumors have presented challenges in predicting tumor behaviors, mainly due to the relatively low numbers of individual tumor types[10].

Understanding biological mechanisms of cancer in patient survival is crucial to support treatment decisions but also to improve cancer prognosis estimates. Several statistical methods have been proposed for modeling survival distributions. For cancer, most popular methods to predict survival times are nonparametric approaches including the Kaplan-Meier estimator[11] and log-rank test[12] or the semi-parametric Cox proportional hazards (Cox-PH) model[13]. While the former two are univariate, the Cox model is a multivariate approach. In its essence, Cox-PH is a multiple linear regression between the event of incidence (which is expressed as a hazard function) and several predictor variables, with the assumption that for each group the hazard of the event is a constant multiple of the hazards in any other group. Importantly, while the Cox model makes parametric assumptions about the predictor variables, it does not make any assumptions about the baseline hazard function itself. Not assuming an underlying distribution of survival times combined with its multivariate nature makes Cox-PH widely accepted as the method of choice to model cancer prognosis and to compare survival characteristics between different groups[14].

The main input sources in current cancer prognosis models are either patient characteristics such as age, gender, tumor stage, and known comorbidities, sometimes in combination with histopathological risk factors[15] or genomic data[16,17]. Recently, more data types are routinely generated and made available via public large-scale collaborative initiatives like The Cancer Genome Atlas (TCGA)[18]. However, as these are now often high-throughput and high dimensional, their complexity and volume present challenges to traditional survival analysis methods. Continuous improvements in deep learning (DL) enabled their use for

complex clinical data and neural networks (NN) have become an increasingly popular tool for survival predictions. Especially the combination of NN architectures with the Cox-PH has gained a lot of momentum, mainly because this approach enables the use of more complex non-linear models on censored data[17]. Two examples are DeepSurv[19] and Cox-nnet[20]. While the former uses patient's clinical characteristics as input features, the latter uses genomic data as input. Both frameworks consist of a NN architecture with a Cox regression model as the output layer, and have shown to outperform traditional Cox-PH models. Interestingly, some studies showed that when these DL models were used on other diseases they also had better results than Cox-PH, suggesting that these models can be transferred and used for different, but conceptually similar tasks[17]. Another example is SALMON that, by also leveraging a combination of NN and Cox-PH, merges clinical data with multi-omics data including mRNA, miRNA, copy number and mutation burden data to estimate breast cancer prognosis[21].

With the technological advances in high-throughput sequencing and digital pathology, more quantitative tumor data is rapidly becoming available. Combined with deep learning algorithms, it is now possible to process this data to its fullest potential. However, while there have been already substantial efforts to develop multimodal data fusion methods for oncology, there are few that combine genomic with histopathological data[22]. Also, so far most of this work relied on the fusion of separately trained models (late fusion)[23]. Joining information from these two data modalities and exploring differences between fusion methods, presents a promising strategy to take advantage of the available heterogeneous information. Ideally, such a unified framework could uncover complex interactions leading to more accurate tumor profiling, ultimately resulting in better survival predictions and better disease management. Recent work in this space indeed showed that a model integrating features from Whole Slide Images (WSI) with genomic data improved prognosis predictions from the unimodal models. For example, Zhan et al. extended the Cox-nnet framework and developed a two-stage Cox-nnet model that integrates pathological images with gene expression data for survival analysis in liver cancer. Their results illustrate that imaging features add additional predictive information, as the combined model is more accurate than the model with gene expression alone[24]. Two other integrative frameworks focusing on renal cell cancer prognosis also had better performance than when using histopathological or genomic features alone[25,26].

Also for brain cancer, a handful of studies investigated the fusion of these complementary modalities for survival prediction. One study trained a convolutional NN (CNN) with Cox-PH as output layer on histopathology images for survival analysis in glioma patients with and without inclusion of two genomic markers, i.e., *IDH* mutation and 1p/19q codeletion[27]. This study showed that the model including the genomic data had better predictive performance than the one without these features. Another example is the "pathomic fusion" framework that uses attention gating and tensor fusion to integrate features from WSIs with genomic features[23]. Validated on both renal and glioma cohorts, this model outperformed unimodal and late fusion algorithms for cancer diagnosis and prognosis. These results show that imaging and genomic data, though very different in nature, both may contain different aspects of disease characteristics that are important for survival prediction.

For adult, but especially for pediatric brain tumor patients, approaches are urgently needed to predict clinical outcome and assist physicians to make treatment decisions. Here, we propose a comprehensive framework to fully integrate multiple data modalities including digital pathology (WSIs) and expression data

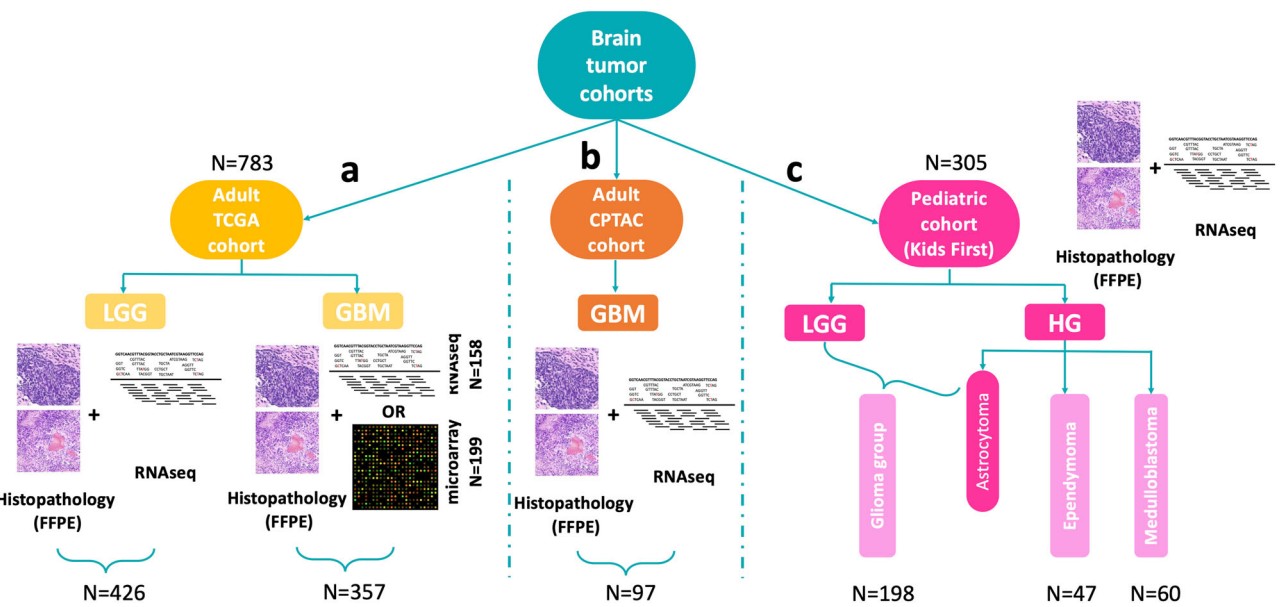

**Fig. 1 Overview of used datasets and samples. a** Adult brain tumor The Cancer Genome Atlas (TCGA) cohort ($N = 783$) consisting of both low-grade glioma (LGG) ($N = 426$) and glioblastoma (GBM) (357) samples. For the LGG samples, both histopathology images and RNA-sequencing data were available. While histopathology images were available for all GBM samples, 158 of these had RNA-sequencing, and 199 microarray expression data. **b** Adult Clinical Proteomic Tumor Analysis Consortium (CPTAC)-GBM cohort with histopathology and RNA-sequencing data ($N = 97$). **c** Pediatric Brain Tumor Atlas (PBTA) Kids First cohort ($N = 305$). This cohorts consists of four tumor subtypes: LGG and high-grade (HG) astrocytoma (combined in one Glioma group, $N = 198$), ependymoma ($N = 47$), and Medulloblastoma ($N = 60$). Both histopathology images and RNA-sequencing data were available for all samples of this pediatric cohort.

(RNA sequencing and microarrays) to develop a unified model for brain tumor survival prediction. The backbone of this framework is composed of different CNN architectures connected to a Cox-PH output module. Using two separate glioma cohorts of 783 adults from TCGA and 305 pediatric patients from Pediatric Brain Tumor Atlas (PBTA), we implement and compare three strategies for data fusion: early fusion, late fusion, and joint fusion. Using a cross-validation approach, our results show that the suggested strategies for multimodal data fusion enable the integrated models to achieve improved prediction accuracy than using histopathology and genomic data in isolation. Interestingly, further analysis of the model interpretation shows that joining pathology features with genomic data adds more meaningful biological insights into specific cancer mechanisms. Additional validation of the adult glioma models on an independent cohort of 97 adult GBM patients illustrates that the multimodal models perform better on new data from different cohorts. Lastly, when transferring the pediatric glioma models to two other pediatric brain tumor subtypes with low sample numbers, all three data fusion strategies perform better. Thus, also in transfer learning an approach using multiple modalities potentially adds value in predicting survival probabilities.

## Materials and methods

**Data**. Brain tumor data from three sources were used: (i) Adult cohort from TCGA consisting of 783 samples[28], (ii) Pediatric cohort totaling 305 samples from the PBTA[29] available through the Gabriella Miller Kids First Data Resource Portal (KF-DRC, https://kidsfirstdrc.org), and (iii) data from 97 adult patients from the National Cancer Institute's Clinical Proteomic Tumor Analysis Consortium Glioblastoma Multiforme (CPTAC-GBM) cohort[30] (Fig. 1).

The used adult brain cancer TCGA cohort consisted of a glioma cohort with LGG and GBM samples. Only samples having both histopathology data and expression data were selected. For

the former data type, TCGA contains imaging data originating from fresh frozen (FF) and formalin-fixed paraffin-embedded (FFPE) tissue slides, but as FFPE fixation better preserves tissue architecture only samples with this data type were included. For a total of 426 LGG samples also RNA-sequencing (RNA-seq) data was available, but only 158 of the GBM samples had corresponding RNA-seq data. To augment this GBM dataset, also samples with microarray expression data instead of RNA-seq were included thereby adding 199 patients. The final adult glioma dataset consisted of 783 samples, with 426 LGG samples and 357 GBM samples (Fig. 1a).

The pediatric brain cancer cohort from PBTA (Kids First) consisted of four tumor subtypes: LGG, HG astrocytoma, HG ependymoma, and HG medulloblastoma. Also for this cohort, only samples having both histopathological images and expression data (RNA-seq) were selected ($N = 305$). LGG and astrocytoma samples were combined in a pediatric glioma group ($N = 198$). For ependymoma and medulloblastoma there were 47 and 60 patient samples, respectively. Compared to glioma, pediatric ependymoma, and medulloblastoma both arise from different cell types and occur in disparate brain regions. As such, these three brain tumor subtypes were considered separate disease entities (Fig. 1c).

The third dataset consisted of 97 GBM patients from the CPTAC cohort having both histopathology images and RNA-seq data. This dataset was not used for training or fine-tuning but kept as an independent cohort for model validation (Fig. 1b).

All data used in this project constitutes secondary data use of publicly available data from The Cancer Genome Atlas project (TCGA), The Children's Brain Tumor Tissue Consortium (CBTTC) project and the National Cancer Institute's Clinical Proteomic Tumor Analysis Consortium Glioblastoma Multiforme (CPTAC-GBM) project. All samples have been collected and utilized following strict human subjects' protection guidelines, informed consent, and IRB review of protocols as part of

these original projects. The relevant data is publicly available for academic research.

## Preprocessing

*Gene expression data.* For both adult and pediatric RNA-seq, genes with missing values were removed after which gene counts were normalized by performing a log-transformation followed by z-score transformation. Microarray data from the TCGA GBM samples was preprocessed in the same way: genes with missing values were removed followed by a log- and z-score transformation. After standardizing the data, the Combat-Seq package was used to account for any batch effects in the adult TCGA expression data[31]. As for the CPTAC and pediatric cohorts all the data came from the same batch, no batch correction step was performed. In the last step the overlapping genes between the TCGA and pediatric cohort were determined (=12,778). Only these were kept in the final expression vector for each sample.

*Histopathology data.* In clinical practice, FFPE or FF specimens are generated and used for histopathological assessment. While FFPE fixation can result in cross-linking, degradation, and fragmentation of DNA, it has several advantages over FF such as preservation of the cellular and architectural morphology, easy storage, and availability[32]. Both have already been used in DL histopathology-focused applications, however, because FFPE better preserves tissue morphology, these images were selected. As in digital pathology a tissue slide is scanned at various magnifications resulting in a giga-pixel WSI, the main challenge of working with digital pathology WSIs is their size. Typically, a WSI is 50k by 50k pixels and does not fit in the memory of a standard GPU. To tackle this high dimensionality, a common workaround is to extract smaller patches at a high resolution[33,34]. After downloading the FFPE images, an OTSU image segmentation was performed to separate target tissue from the background[35]. Non-overlapping patches of $224 \times 224$ pixels were subsequently extracted from the foreground region at a ×20 resolution using the OpenSlide library[36]. In case multiple slides were available for the same patient, patches were extracted from all slides. The median number of patches for a WSI was 2900 for the adult cohort and 6600 for the pediatric cohort. One important factor when dealing with clinical images is the presence of stains (mainly from hematoxylin and eosin) buts also the variability originating from different hospitals[37]. Stain normalization and/or augmentation steps are essential steps for more robust and better-performing models, with the latter shown to be the most important for CNNs with the limited added value of the former[38]. For this reason and the fact that color normalization comes with computational challenges, it was chosen to perform a stain augmentation step by adding random color jitter (using ColorJitter function from PyTorch) to mask the hematoxylin and eosin stains without normalization.

## Survival prediction

The goal of survival prediction is to predict the likelihood that a patient will survive till time $t$. As already described above, Cox-PH is a widely used method to model patient's survival times given their baseline feature data $X$. Cox-PH model is expressed by the hazard function $h(t)$ which can be estimated for a patient $i$ as:

$$h(t|X_i) = h_0(t)e^{(\beta X_i)} \quad (1)$$

with $h_0(t)$ the baseline hazard function, $X_i$ the patient's covariate vector, $\beta$ the corresponding coefficient vector, and $\beta X_i$ the log-risk function.

One major advantage of this model is that it can deal with censored data, i.e. data where some of the patients have an unknown time of event (death) due to missing data or the fact that they are still alive at the end of the study. To estimate the $\beta$ values in the Cox-PH model, a partial likelihood function is constructed and optimized to get a maximum likelihood estimator without the need to estimate or know the baseline hazard $h_0$[13]. Thus, when performing a Cox regression, the coefficients vector $\beta$ is tuned to optimize the partial likelihood, which in practice is done by minimizing the negative log-likelihood, also known as Cox loss:

$$\mathcal{L}(\beta|X) = -\sum_{i|C_i=1}\left(X_i\beta - \log\left(\sum_{j|Y_j \geq Y_i} e^{X_j\beta}\right)\right) \quad (2)$$

with $\beta$ the coefficient vector *and* $X_i$ the covariate or feature vector, $Y_i$ the survival time and $C_i$ the censor indicator for patient $i$.

Importantly, this loss function can be adapted and implemented in a DL architecture to enable more complex non-linear relations. In the Cox module of our framework, $X_i\beta$ is changed by $f_\theta(X_i)$ where $f_\theta$ represents a non-linear mapping learned by the first layers of our NN that extract the patient features from the input data. Here, $\theta$ constitutes the model parameters including weights and biases of each NN layer. Our objective is thus to minimize the following loss function:

$$\mathcal{L}(\beta|X) = -\sum_{i|C_i=1}\left(f_\theta(X_i) - \log\left(\sum_{j|Y_j \geq Y_i} e^{f_\theta(X_i)}\right)\right) \quad (3)$$

This Cox module is used as the output layer of our NN framework and generates survival scores for each sample. Note that in practice $\sum_{j|Y_j \geq Y_i} e^{f_\theta(X_i)}$ is not determined for all patients, but a batch sampling strategy is used where this sum is calculated for the patients of each current batch.

## Feature extraction

*Feature extraction and survival prediction for histopathological data.* To extract histopathology features from the tissue slides, the generated $224 \times 224$ patches are used as input of a ResNet-50 CNN[39] (see section Model Training for more details) which returns a feature vector of size $2048 \times 1$ for each patch (Fig. 2a). These features are next mapped to a survival (risk) score through a fully connected output layer using the adapted Cox loss (Equation 3) as loss function. Model training was patch-based, i.e., the model aims at predicting a survival score for each patch. During the model evaluation, the risk scores of all patches from the same patient are averaged to obtain one final patient's risk score. Similarly, when needed for downstream analysis, features of all patches from the same patient are averaged to derive one global feature representation for each patient.

*Feature extraction and survival prediction for expression data.* Feature extraction of the high dimensional expression data was done using a multi-layer perceptron (MLP) architecture (see section Model Training for more details). Using the same approach as for the histopathology data, this MLP outputs for each sample a genomic feature vector of size $2048 \times 1$ (Fig. 2b) which is then mapped to a risk score through a fully connected output layer using the adapted Cox loss (Equation 3) as loss function.

**Multiscale data fusion.** Although every single modality may be predictive by itself, the aim of this study is to explore the added value in predictive power when these modalities are combined in one model. Because histopathological imaging data and genomic data are both highly dimensional and cover different scales, it is not feasible to directly combine the data at the input level. Here,

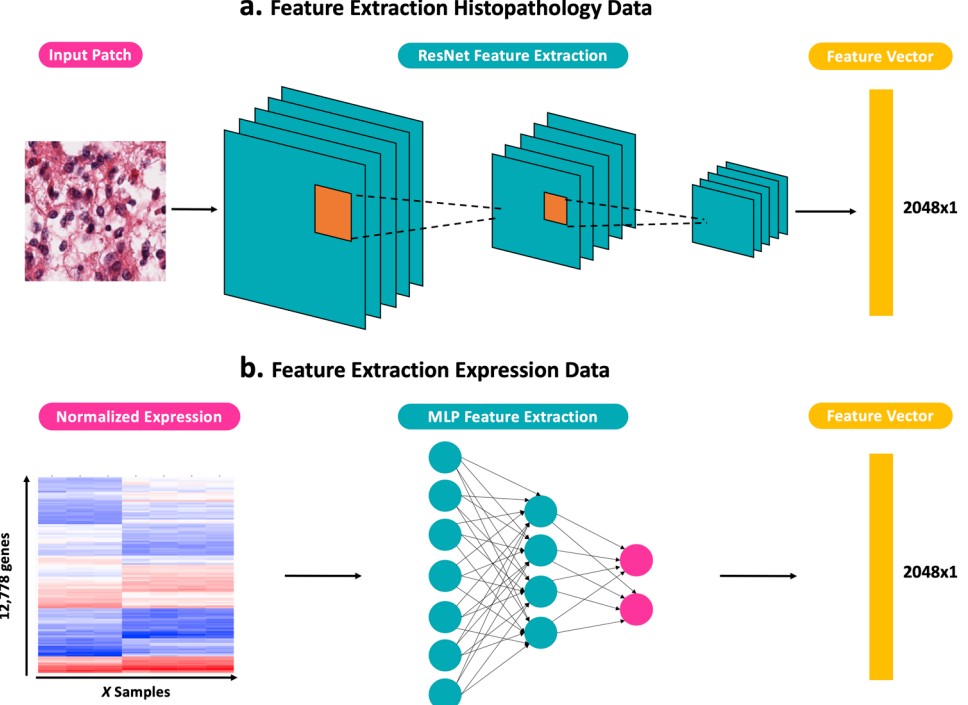

**Fig. 2 Feature extraction and survival prediction for histopathological and expression data. a** ResNet-50 feature extraction flow for pathology image patches. **b** Multi-Layer Perceptron (MLP) feature extraction flow for gene expression data. Both extraction flows produce a modality-specific feature vector of dimension 2048 × 1.

three data fusion strategies are examined to integrate these two data types: (i) early fusion (ii) late fusion, and (iii) joint fusion.

**Early fusion**. The first approach, also known as feature fusion, is characterized by concatenating the features extracted from the imaging data, i.e. the 2048 × 1 feature vector outputted by the Resnet-50, with the extracted genomic features, i.e., the 2048 × 1 feature vector generated by the MLP. For the imaging data, feature vectors of all used patches from the same patient are first averaged to derive one global feature representation for each patient. After concatenating these global feature vectors with the genomic feature vectors for all patients, a new Cox module (MLP) is trained on this merged data with the adapted Cox loss as loss function. Note that in this setup, the two feature extraction models as well as the final MLP are independently trained with distinct Cox modules (Fig. 3a). The final target value is the risk score generated by the last Cox module.

**Late fusion**. In late fusion, for each data modality a separate model is trained, and fusion is done at the output stage. In other words, the risk scores predicted by each independent model are combined and a new Cox regression model (Cox module) is trained on these merged scores. Also here, each model is trained independently, and the final output value is the risk score estimated by the last Cox module (Fig. 3b).

**Joint fusion**. The last strategy is a joint data fusion approach where the histopathology and expression modalities are trained simultaneously to generate a joint feature vector that is then fed into a fully connected layer optimizing for Cox loss. Thus, in this intermediate approach, the feature extractions from the histopathology images and the expression data are learned at the same time and together with the final risk score. In contrast to the previous two strategies, here the Cox loss is fed back to the feature extraction layers thereby affecting feature learning (Fig. 3c).

## Model development

*Model training*. For both the adult TCGA and pediatric glioma cohorts five models were trained: (i) histopathology model, (ii) RNA expression model, (iii) early fusion multimodal model, (iv) late fusion multimodal model, and (v) joint fusion multimodal model. For each cohort, samples were shuffled into a training and test set at an 80/20 ratio with stratification on age, gender, tumor grade (high/low), and survival time. This test set was left out during model training and only used for calculating model performance. A 10-fold stratified cross-validation (CV) was performed on the training set of the adult cohort, while a fivefold stratified CV strategy was chosen for the pediatric glioma cohort (since this cohort contained fewer samples). The optimal weights for each model were saved based on the epoch that achieved the highest validation accuracy. The best model was chosen based on the CV configuration with the lowest validation loss. This final model was next evaluated on the test set.

For the histopathology model a ResNet-50 CNN architecture[39] was chosen with the ADAM optimizer[40]. Weight initialization was done using the weights of a pretrained model on ImageNet[41]. After empirical testing, only the last ResNet block was further fine-tuned while freezing the other blocks, as fine-tuning more blocks resulted in overfitting the training set. Following a grid search an optimal learning rate of 5e-4 was set for the adult cohort and 5e-3 for the pediatric cohort. The model was trained using a batch-size of 128 patches and for each batch the associated loss was the Cox loss between the patches of that batch. Note that in this setup it is not possible to directly optimize the loss for all patients, but that the model learns to distinguish high- from low-risk patients within a batch. However, using a higher batch-size of 128 makes this approximation robust. To increase computational efficiency, not all patient patches are used for training but for each WSI 100 random patches are selected. This number was chosen after comparing performance versus computational resources and time using either 1, 10, 100, 500,

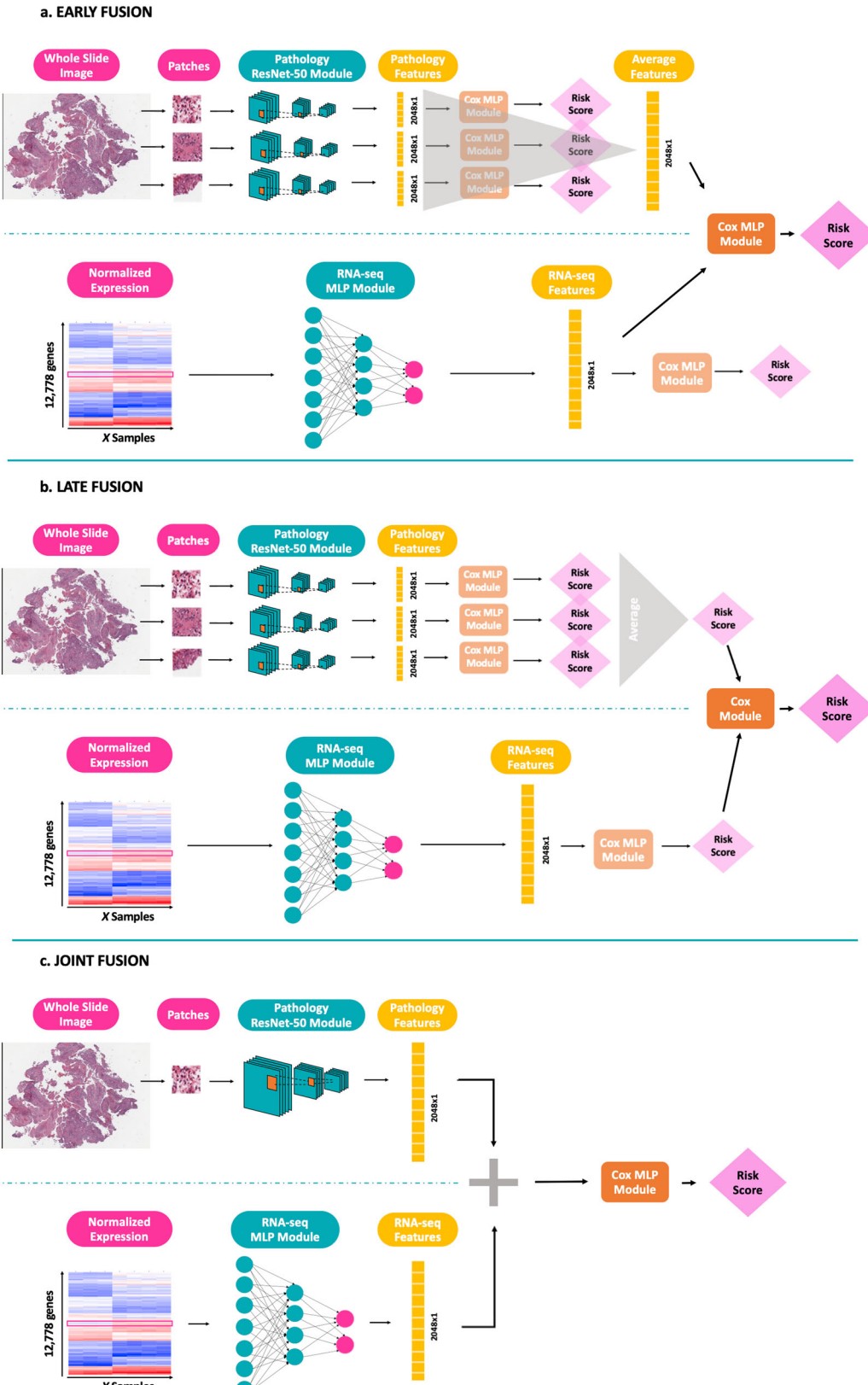

**Fig. 3 Visualization of the three data fusion strategies to integrate histopathological and expression data. a** Early or feature fusion. **b** Late fusion, and **c** Joint fusion.

1000, or 2000 patches. For each number, 10 training runs were performed and CI scores were calculated for the test set (Supplementary Figure S1). After this experiment a total of 100 random patches per WSI was chosen as a trade-off between performance and computational complexity.

For the RNA expression model, a three-layer MLP was used with an input layer of size 12,778 (= number of genes), hidden layer of size 4096, and output layer with a dimension of 2048. Before each layer, a dropout with probability 0.5 was implemented, and the non-linear ReLu function was used as activation function. The 2048 × 1 feature vector is next mapped to a risk score through a fully connected output layer (Cox MLP module) with an output dimension of size 1 (the risk score) using the adapted Cox loss as loss function. The ADAM optimizer was also used for this RNA model. Using the same approach as for the histopathology model, an optimal learning of 1e-5 was applied for the adult cohort, and 1e-4 for the pediatric cohort.

As described above for the early/feature fusion multimodal model, after concatenating the imaging features and genomic features into a 4096 × 1 vector this data is used as input data of a MLP with two hidden layers. The adapted Cox loss is again used as loss function together with the ADAM optimizer. Similar as for the RNA model, a dropout with probability 0.5 was implemented before each layer, and the non-linear ReLu function was used as activation function. This early Cox MLP module has an input layer of size 4096, two hidden layers of size 2048 and 200 and output layer of size 1. For both the adult and pediatric cohort, a learning rate of 1e-6 was applied.

The joint fusion model consists of a histopathology model and an expression model that are trained simultaneously and are then connected to a fully connected output layer (Fig. 3c). A potential issue that comes with this joint training approach is that the learning rates of the two modalities might differ. As histopathology training is patch-based, the model needs to go through all patches of a patient to learn the patient's feature representation. For the expression data, on the other hand, each generated feature vector immediately corresponds to one patient. For this reason, the genomic modality may learn a patient's expression features faster than the histopathology modality. To tackle this potential discrepancy and to balance the learning speed, for each submodule different learning rates are used, with a higher learning rate for the histopathology feature extraction (5e-5 for the adult cohort and 5e-4 for the pediatric cohort) and a smaller learning rate for the genomic feature extraction (1e-6 for the adult cohort and 1e-5 for the pediatric cohort). As this joint strategy is the most complex model, a dropout with probability 0.8 is added to the final output layer together with a larger learning rate (1e-2 for the adult cohort and 1e-3 for the pediatric cohort) to avoid overfitting.

*Performance metrics.* Model performances on survival prediction were evaluated with two standard evaluation metrics: (i) the Concordance Index (CI)[42,43] and (ii) the Integrated Brier Score (IBS)[44]. The CI is a performance measure that evaluates how well the predicted risk score ranks patients according to their actual survival time. It is calculated by dividing the number of pairs of subjects whose predicted risks are correctly ordered, by the number of admissible pairs of subjects. As such, a value of 1 indicates perfect prediction with all pairs correctly ordered, and a value of 0.5 indicates random prediction. In this formula, a pair of subjects is considered as admissible if none of the events in the pair is censored or the earlier time in the pair is not censored. Another metric, the Brier score, is calculated by the squared differences between observed survival status and the predicted survival probability at a given time point. The IBS represents the overall Brier score for all available times and evaluates the

accuracy of the survival predictions. Contrary to the CI, an IBS of 0 indicates perfect predictions, while an IBS of 1 indicates entirely inaccurate predictions. IBS values were calculated with the "survcomp" package in R[45].

Here, we defined a "Composite Score" (CS, Equation 4) that consists of both the CI and IBS with the aim to combine these two scores, each with different purposes, in one single metric:

$$CS_{modelX} = \frac{CI_{modelX} + (1 - IBS_{modelX})}{2} \qquad (4)$$

The CS is thus an average of a relative measure (CI) and an absolute error metric (IBS) and allows straightforward evaluation and comparison of different models. Note that in this definition, a value of 1 indicates perfect prediction, while a value of 0 implies entirely inaccurate predictions.

Kaplan-Meier curves are used as a final evaluation method to visualize survival predictions and performance of each model[46]. To generate these curves, test data were first divided into two groups: (i) a poor survival group containing samples with predicted risk scores greater than the median predicted risk, and (ii) a good survival group, containing samples with predicted risk scores smaller than the median predicted risk. Next, Kaplan-Meier curves are generated for these two groups in one figure per model.

**Model validation.** To evaluate the adult TCGA glioma models, additional validation was done on an independent dataset of 97 GBM adult patients from the CPTAC repository. This dataset was not used in any training or fine-tuning step but kept as a separate independent cohort for final model validation.

**Transfer learning.** The technique transfer learning enables NNs trained on one task to be repurposed to another related task. The knowledge learned from the previous training round is transferred to the current task and is often done to save time, get better performance, or when fewer data are available for the second task. Here, we explored if information learned from one brain tumor cohort can be leveraged to make survival predictions on another cohort. Specifically, we investigated transfer learning of the pediatric glioma models to the other two pediatric brain tumor subtypes, i.e. ependymoma and medulloblastoma. Although each of these brain tumor cohorts—pediatric glioma, pediatric ependymomas, and pediatric medulloblastoma—is characterized by different clinical features, such as patient demographics, brain regions, tumor subtypes/cells of origin, they may share some known or unknown commonalities that could be exploited. A direct transfer of the pediatric glioma models was done on the smaller ependymoma and medulloblastoma cohorts and survival predictions were assessed.

**Model Interpretability analysis.** To compare unimodal interpretability versus multimodal interpretability of the expression data, the relation between the RNA input features and the models' predictions was assessed by backpropagation of the gene expression (RNA only) and joint fusion models. For each of the 12,778 genes, the gradient (= the relative importance with respect to the survival prediction) was calculated for all cohort samples. Using the Molecular Signature Database (MSigDB) from Broad Institute, individual genes were next mapped to the Reactome pathway collection (C2:CP collection v7.5, downloaded from https://data.broadinstitute.org/gsea-msigdb/msigdb/release/7.5/)[47,48]. The importance of each pathway was assessed by averaging the gene gradients of the associated gene set. Here, negative gradients indicate contribution of the pathway to a lower risk prediction and positive

**Table 1 Composite Scores (CS) of survival predictions on adult and pediatric glioma cohorts using two single modality models (FFPE & RNA), and three data fusion methods (early, late & joint).**

| Cohort | Strategy | Model | Training set CV ean(CS) ± stdev | Validation set CV Mean(CS) ± stdev | Test set CS |
|--------|----------|-------|-------------------------------|-----------------------------------|-------------|
| Adult | Single modality | FFPE | 0.874 ± 0.016 | 0.807 ± 0.018 | 0.805 |
| | | RNA | 0.861 ± 0.022 | 0.809 ± 0.037 | 0.780 |
| | Multimodal | Early Fusion | 0.876 ± 0.008 | 0.831 ± 0.022 | 0.836 |
| | | Late Fusion | 0.891 ± 0.011 | 0.832 ± 0.019 | 0.822 |
| | | Joint Fusion | 0.876 ± 0.026 | 0.824 ± 0.019 | 0.822 |
| Pediatric | Single modality | FFPE | 0.900± 0.010 | 0.792 ± 0.070 | 0.854 |
| | | RNA | 0.900 ± 0.050 | 0.907 ± 0.034 | 0.811 |
| | Multimodal | Early Fusion | 0.895 ± 0.037 | 0.880 ± 0.028 | 0.919 |
| | | Late Fusion | 0.930 ± 0.032 | 0.877 ± 0.059 | 0.913 |
| | | Joint Fusion | 0.981 ± 0.005 | 0.911 ± 0.022 | 0.883 |

The first two data columns show the results (mean and standard deviation (stdev)) on the cross-validation (CV) training and validation sets (10-fold CV for adults and fivefold CV for pediatric cohort). The last column shows the result of the final model on the test set.

gradients indicate contribution to a higher risk prediction. In a final step, these pathway gradients were visualized in a summary plot using SHAP (SHapley Additive exPlanations)[49].

Similarly, to investigate model interpretability of the histopathology data, both a qualitative and quantitative analysis was performed on the FFPE only and joint fusion model predictions. For the former, predicted risk scores of the test set samples were backpropagated and model gradients were determined for the input patches. For each patch, model features were next visualized by plotting a saliency map that highlights absolute gradient values (model importance) for each pixel. The resulting heatmap is next overlaid with the original patch (transparency value of 0.5) to visualize the tissue regions which contributed the most to the outputted risk sore. In addition to model visualization, a quantitative analysis was performed by segmenting the different cell types within the input patches using a HoverNet network pretrained on the PanNuke dataset[50,51]. Cell type distributions were compared between bad and good survival samples as well as between high-risk and low-risk patches in one sample.

**Reporting summary**. Further information on research design is available in the Nature Portfolio Reporting Summary linked to this article.

## Results

**Multimodal data fusion and model development on glioma cohorts**. A total of five models (two single and three multimodal models) were developed and evaluated for both the adult TCGA and the pediatric glioma cohorts. Resulting composite scores are shown in Table 1, corresponding CI and IBS scores can be found in Supplementary Tables S1 and S2. The first two columns show the mean score with the standard deviation of each model, while the last column shows the performance of the best-performing CV model on the test set. Figure 4 displays the corresponding Kaplan–Meier curves for the test set of the adult cohort (left panel) and the test set of the pediatric cohort (right panel).

For the adult glioma data, the Kaplan–Meier curves of all five scenarios show large separations for the poor survival group and good survival samples of the test set (log-rank test *P* value <0.0001, *N* = 156). Supplementary Table S3 shows the distribution of different diagnostic and genetic subtypes amongst these poor survival and good survival groups. The histopathology and gene expression single modality models achieve good survival predictions, with a CS of 0.805 and 0.780 on the test set, respectively. The data fusion models have better performances than single modality models, achieving CS scores of 0.822 for the late and feature fusion models and 0.836 for the early fusion

model. A pairwise comparison (Wilcoxon signed-rank test for 1:1 comparison and Kruskal–Wallis test followed by Dunn's test for multiple pairwise comparison) of the CS distributions for the validation and test sets indeed revealed significantly improved predictive performance of the multimodal models over the unimodal models (Fig. 5, Supplementary Table S4).

Next, for the pediatric glioma data the Kaplan–Meier curves show a good split between the high and low-risk patients of the test set (log-rank test *P* value <0.03, *N* = 39), with the best separation for the multimodal fusion models. While also for this cohort, the unimodal models achieve good performance (CS of 0.854 and 0.811 respectively for the histopathology and gene expression model), the multimodal models again result in better survival predictions for the test set samples with the best performance for the early fusion model (CS of 0.919). Boxplots of the CS distributions for the validation and test sets are shown in Supplementary Figure S2. Note that due to the limited sample size (fivefold CV) it was not possible to perform a Wilcoxon or Kruskal–Wallis pairwise comparison.

In summary, we observe improved predictive performance in overall survival prediction from data fusion methods on both adult and pediatric multi-scale data cohorts, with the early fusion strategy the best-performing model in our experiments. These results suggest that integration may have discovered additional information that is not revealed in either modality when used in isolation.

**Model evaluation on CPTAC**. After preprocessing the pathology images and expression data of the CPTAC samples, the developed adult glioma models were next validated on this external cohort. As this cohort only consists of GBM samples, especially the IBS is of importance here while the CI and Kaplan–Meier curves less relevant for this more homogeneous cohort (all HG). The CS and IBS of each model can be found in Table 2, CI scores in Supplementary Table S5. The multimodal models have a slightly better CS compared to the histopathology and expression models. Also in terms of IBS, fusion of the two modalities improves the predictions with early fusion as the best scoring strategy.

**Transfer learning between pediatric brain tumor subtypes**. Next, transfer learning was performed from pediatric glioma to two other pediatric brain tumor subtypes with fewer available samples, i.e., ependymoma, and medulloblastoma. Note that like the CPTAC cohort, these two cohorts only consist of high-grade samples. As such, also in this case the IBS is of special interest. Table 3 contains both the CS and IBS performance of each model with CI scores reported in Supplementary Table S4.

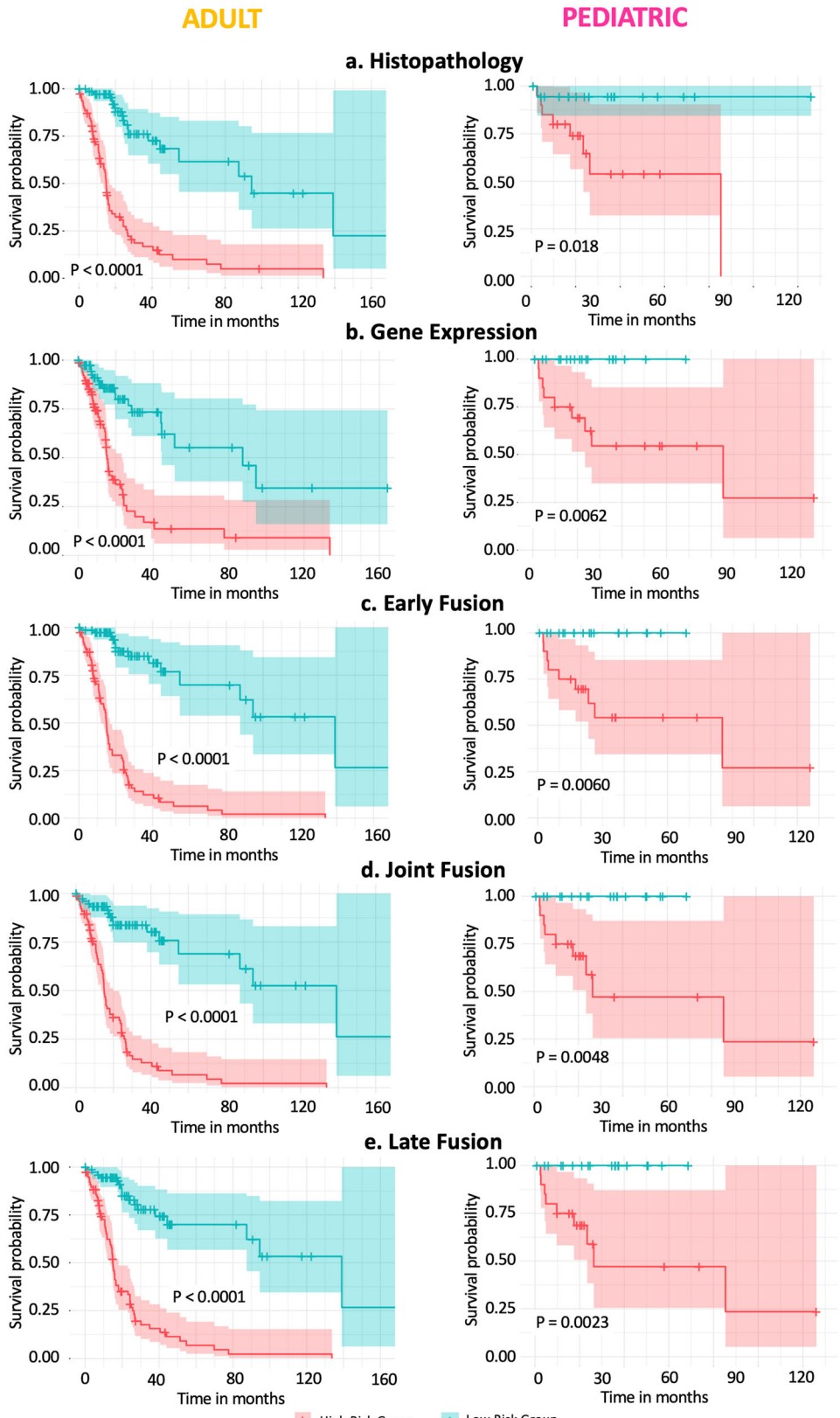

**Fig. 4 Kaplan–Meier curves of the adult glioma (N = 156) and pediatric glioma (N = 39) test set. a** Histopathology model. **b** Gene expression model. **c** Early or Feature fusion model. **d** Joint fusion model. **e** Late fusion model.

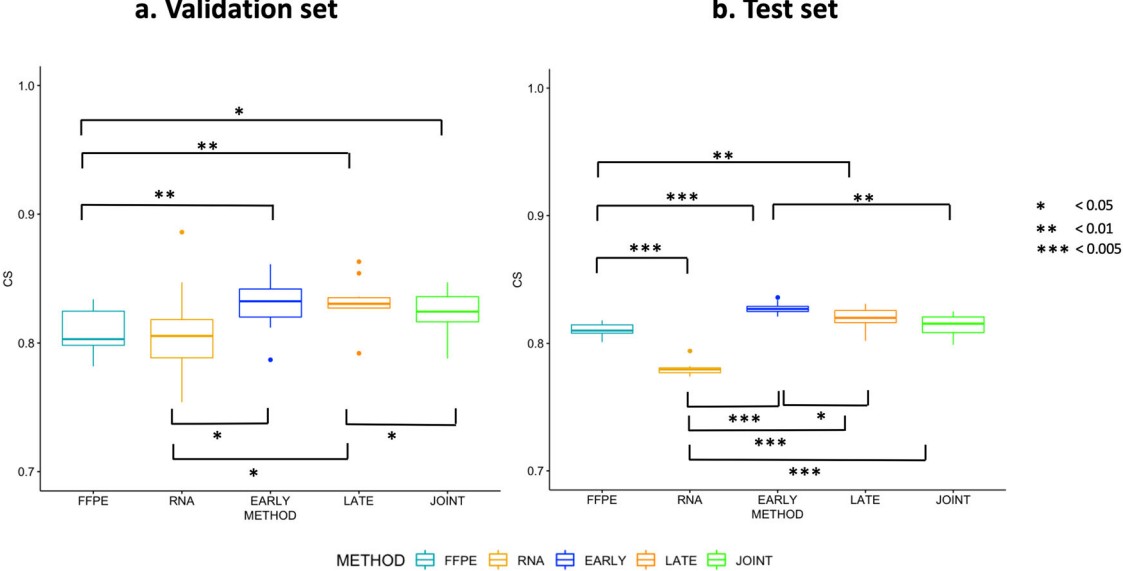

**Fig. 5 Boxplots of model performance for each model strategy on the adult glioma cohort. a** Composite Score (CS) distribution on cross-validation (CV) validation sets (N = 63). **b** CS distribution of each CV fold on the test set (N = 156). (*P value <0.05, **P value <0.01 and ***P value <0.005; pairwise Wilcoxon signed-rank test).

**Table 2 Results of survival predictions on the independent Clinical Proteomic Tumor Analysis Consortium (CPTAC) cohort using the models trained on the adult The Cancer Genome Atlas (TCGA) cohort.**

| Strategy | Model | CPTAC | |
|---|---|---|---|
| | | CS | IBS |
| Single modality | FFPE | 0.663 | 0.237 |
| | RNA | 0.679 | 0.205 |
| Multimodal | Early Fusion | 0.710 | 0.174 |
| | Late Fusion | 0.696 | 0.195 |
| | Joint Fusion | 0.688 | 0.194 |

As this CPTAC cohort only consists of a homogeneous glioblastoma (GBM) high-grade (HG) population, both the Composite Score (CS) and Integrated Brier Score (IBS) are shown.

**Table 3 Transfer learning results of pediatric glioma models on ependymoma and medulloblastoma pediatric cohorts.**

| Strategy | Model | Ependymoma | | Medulloblastoma | |
|---|---|---|---|---|---|
| | | CS | IBS | CS | IBS |
| Single modality | FFPE | 0.601 | 0.318 | 0.544 | 0.407 |
| | RNA | 0.670 | 0.290 | 0.683 | 0.251 |
| Multimodal | Early Fusion | 0.701 | 0.187 | 0.710 | 0.218 |
| | Late Fusion | 0.706 | 0.158 | 0.661 | 0.286 |
| | Joint Fusion | 0.751 | 0.114 | 0.680 | 0.336 |

As these two other cohorts consist of a homogeneous high-grade population, both the Composite Score (CS) and Integrated Brier Score (IBS) are shown.

For both cohorts, direct transfer of the glioma histopathology model has the lowest performance for all metrics. The glioma expression model on the other hand produces good survival predictions. However, using the multimodal glioma models seems to boost performance, especially in terms of IBS. For ependymoma, all data fusion strategies perform better with the joint fusion approach having the best score. For medulloblastoma, only early fusion of the features boosts performance over the RNA

expression model alone. Thus, also when using transfer learning the two modalities add value in predicting survival probabilities.

**Model interpretation**. To interpret the genomic part of the models and assess if they detect any relevant biological networks, a pathway analysis was performed on the survival predictions by the joint fusion model and the gene expression model. More specifically, for each of these two models, the relative importance of the gene expression features with respect to the survival predictions, i.e., the gene gradients, were assessed by backpropagation of the predicted risks. Individual genes were next grouped into Reactome pathways and average pathway gradients were calculated. Figure 6 visualizes the average gradients of the top 15 pathways that impact the survival prediction of the pediatric glioma cohort, with negative gradients indicating a contribution to a lower risk score for the sample, while positive gradients lead to a higher risk score. Supplementary Figure S3 shows the same visualization for the adult TCGA glioma cohort. For both models, known cancer pathways are detected. Note that the joint model correctly identifies poor prognosis pathways (positive values) related to *RAS* and *NTKR* signaling and that the good prognosis pathways (negative values) capture pro-apoptotic pathway and *RUNX3*, which was recently reported as a tumor suppressor gene for glioma[52]. These results disappear in the RNA only model, suggesting that the inclusion of histopathology images helps the model select causal genes. On the other hand, while using expression data alone many gene signatures are identified that are predictive of survival, but few are biologically relevant or are more upstream of the causal pathway.

A qualitative assessment of the histopathology (FFPE only) and joint fusion model was performed by visualizing model gradients on the input patches. Figure 7 shows the original tissue patch and the resulting saliency maps for two samples of the adult glioma test set, one bad survival sample with high-risk model score (Fig. 7a) and one good survival sample with low-risk model score (Fig. 7b). For both samples, a patch with a high-risk score and a patch with a low-risk score in the two models is shown. A fourth panel also displays the predominant cell types within each patch determined by HoverNet[50]. For the bad survival sample (Fig. 7a), it can be seen that for the patch with a higher risk score, both

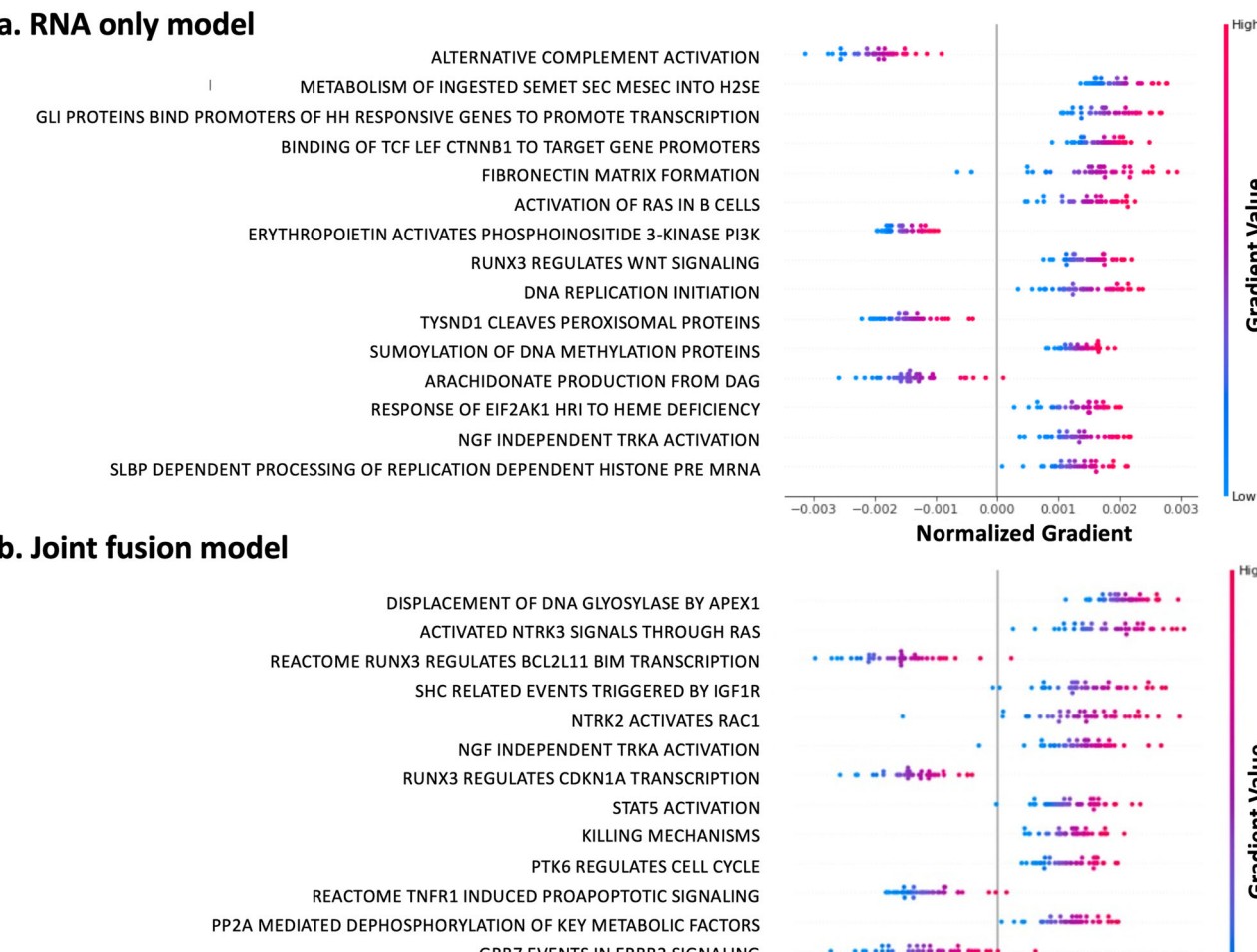

**Fig. 6 Visualization of pathway importance with respect to survival predictions for the pediatric glioma cohort.** Pathways are ranked from top to bottom based on the sum of the absolute gradients across all samples. Negative gradients contribute to a lower risk score, while positive gradients lead to a higher risk score. **a** Top 15 pathways of the unimodal gene expression model (RNA only). **b** Top 15 pathways of the multimodal joint fusion model (histopathology + RNA data).

models focus on the tumor cell region in the upper right. Interestingly, for the patch where both models predict a lower risk score, the higher gradients of the joint fusion model predominantly correspond to the regions with lymphocytes, while for the FFPE-only model higher gradients overlap with a region containing lymphocytes, tumor cells, and epithelial cells. For the good survival sample (Fig. 7b), the regions of interest in the high- and low-risk patches seem to overlap for both models, but with some differences in the location of the highest intensities (gradient values). In this example, important regions correspond with lymphocytes, dead cells, and tumor cells, as well as some epithelial cells. Figure 7c shows the fraction of cell types in samples of the adult glioma test set with a high-risk score (bad survival) and low-risk score (good survival) in both the joint fusion and histopathology model. The clearest distinguishing cell types between these two groups are tumor cells and lymphocytes with a higher fraction of tumor cells in high-risk samples (two sample $t$ test $P$ value <2.2e-16, $t = 22.682$, $df = 16679$, high-risk $N = 78$, low-risk $N = 78$) and a higher fraction of lymphocytes in low-risk samples (two sample $t$ test $P$ value <2.2e-16, $t = -43.225$, $df = 16679$, high-risk $N = 78$, low-risk $N = 78$).

## Discussion

Using single data modalities to build predictive models for multifactorial diseases such as cancer might not provide sufficient information about the heterogeneity of the disease to improve clinical decision making. Development of effective biomedical data fusion approaches is becoming increasingly important as combining multi-scale biomarkers can be more accurate than any single modality alone[53]. While most cancer biomarkers are based on molecular data such as gene expression, histopathology images potentially harness complementary information about the cellular and morphological architecture of the tumor. Here, we propose a framework to integrate these two data modalities to build a unified prediction model for brain tumor survival analysis.

For an adult glioma cohort and a pediatric glioma cohort, five independent survival prediction models were developed: one for each single data modality and three multimodal models based on different data integration strategies: early fusion, late fusion, and joint training fusion. Models were evaluated using multiple metrics: IBS, CI, and Kaplan-Meier curves. While the latter is used to make a visual assessment of the predicted survival curves for high-risk and low-risk samples, the former two provide a

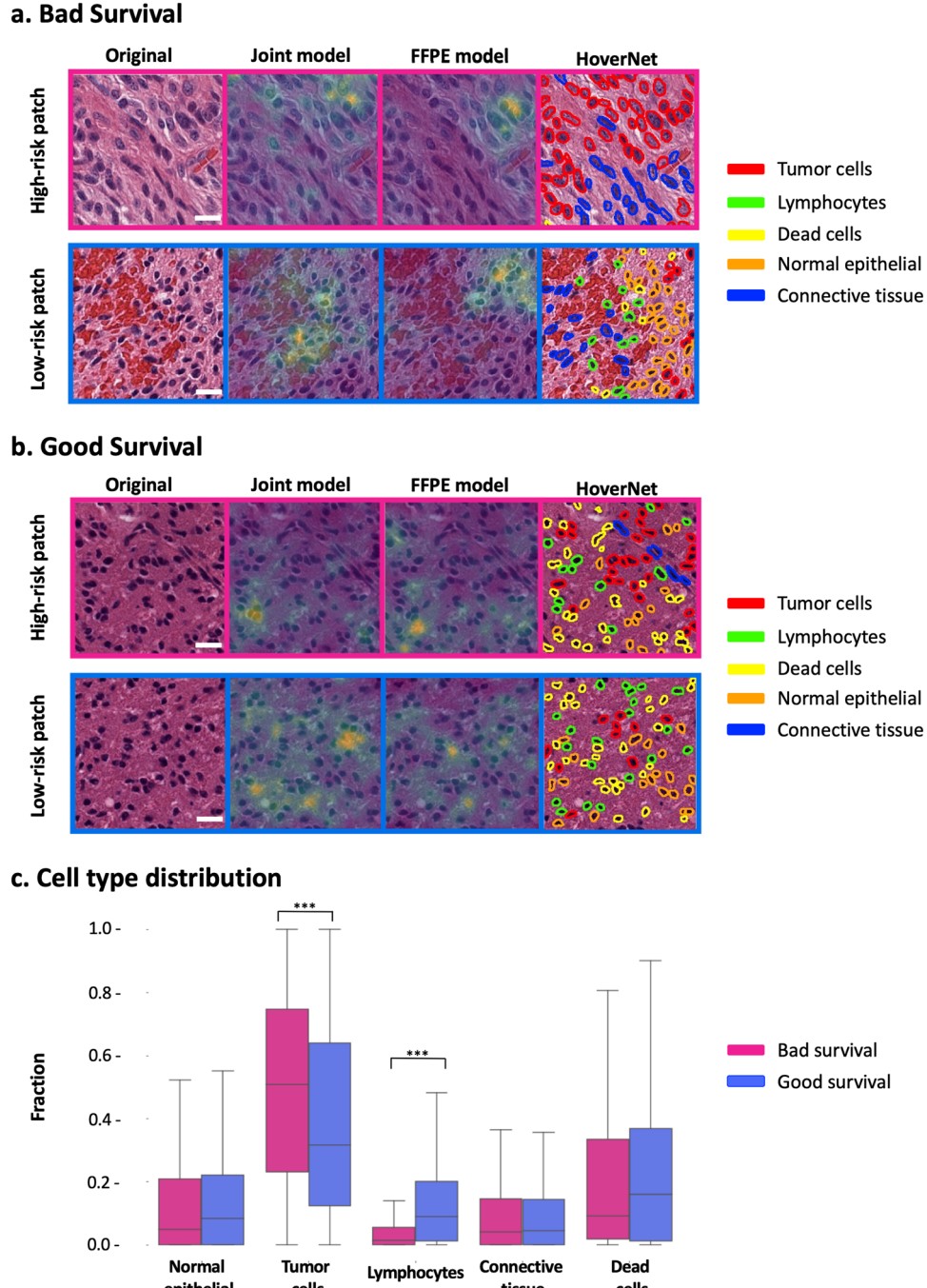

**Fig. 7 Interpretability analysis of histopathology model with respect to survival predictions and cell type distributions. a** Examples of high-risk and low-risk patches for a bad survival sample. **b** Examples of high-risk and low-risk patches for a good survival sample. Left panel = original 224 × 224 patch, scale bars (white insets), 20 µm; middle panels = overlayed saliency map for joint fusion and histopathology (FFPE only) models, Right panel = cell segmentation predicted by HoverNet[50]. **c** Quantitative analysis of cell type composition in extracted 224 × 224 patches of bad survival and good survival samples of the adult glioma test set. (***P value < 2.2e-16, two sample t test, high-risk N = 78, low-risk N = 78).

numeric performance value. Where the IBS is an absolute error metric indicating overall model performance, the CI represents a relative measure rating the discriminative ability of a model. For a more straightforward evaluation and comparison between the different models, a composite score (CS) was defined. This score ranges from 0 to 1, with 1 for perfect predictions, and merges the CI and IBS scores in one combined metric. Using this CS as main performance metric, the proposed fusion strategies achieve synergistic performances and result in significantly better survival predictions on the test set than using each single modality alone.

For both cohorts, the early fusion model attains the best performance with a 0.4 and 0.7 increase in performance for adult and pediatric test set, respectively, compared to the CS of the best-performing single data model, FFPE. When testing our adult glioma models on a third independent GBM dataset, the multimodal models perform better as well, with the early fusion again being the best model. Interestingly, while the model trained on histopathology data alone obtained pretty good performance on the training cohort, it's the least performing model when used on this external cohort. This hints that multimodal models might

be less cohort-specific while single data models are more prone to overfitting on a specific training cohort. Indeed, as multimodal approaches are intrinsically a type of ensemble learning, they can have an advantage when one of the single modalities is not accurately captured in external cohorts.

Many DL characteristics such as unsupervised learning and the capability of handling complex data have increased its implementation in cancer research[54]. However, one major limitation is the need for large datasets which puts constraints on clinical applications with small sample sizes. Transfer learning can circumvent this drawback by leveraging a pretrained model for a different but similar task[55]. With this in mind, we explored whether information learned from the pediatric glioma cohort could be used to predict prognosis on two other pediatric brain tumors for which samples are sparser, i.e. ependymoma and medulloblastoma. Our results indeed show that pediatric glioma models can be directly used for survival prediction on the other subtypes with reasonable performance. Also, the multimodal models perform better than either single data model. In line with the previous results, the early fusion model produces good scores for both ependymoma and medulloblastoma, with the joint fusion strategy the best-performing model for ependymoma. The effectiveness of transfer learning demonstrated in our framework is potentially helpful for survival predictions in a more general context, where there is heterogeneity in the patient cohorts, as well as in the context of pediatric brain tumor subtypes with small cohort sizes where it is intractable to develop de novo models.

Although NNs have the reputation to be "black boxes", techniques exist to make (part of) these models interpretable. Here, we combined saliency maps with cell segmentation analysis to visualize which histopathology features (tissue morphologies) are important in calculating the risk score in both the histopathology and joint model. Post-hoc quantification of cell type composition in histopathology patches showed a difference in tumor cell and lymphocyte fraction between predicted bad and good survival samples. In glioma, lymphocyte infiltration around the tumor has indeed already been associated with a better prognosis[56]. We also investigated how and which of the gene expression features are related to the model output by backpropagation of the predicted risk scores and performing a pathway analysis for each sample. For both the gene expression model and the joint model, known cancer pathways are found including cell cycle pathways linked to poor prognosis, and pro-apoptotic signaling related to good prognosis. For example, *PTK6*-directed cell cycle activity and *NTRK2/3* activation through *RAS*, *RAC1,* and *CDK5* pathways are identified as poor prognosis pathways consistent with previous reports[57–59]. Contribution of *RUNX2* and *RUNX3* expression is also observed. *RUNX* genes are frequently deregulated in various human cancers, indicating their prominent roles in cancer pathogenesis including glioma[52,60]. *RUNX2* is upregulated in multiple cancers and is also found to contribute to glial tumor malignancy[60]. Elevated *RUNX3* expression has also been observed in various metastatic cancers, such as leukemia[61], but in glioma it has been shown to be a tumor suppressor[52], and in our results, these pathways are enriched in good prognosis. Growing evidence indeed indicates that the roles of *RUNX* genes in carcinogenesis are cell type-specific and context-dependent. Interestingly, while signaling and regulation of *RUNX* transcription factors are amongst the top pathways of the joint fusion model, these observations are not prevalent in the RNA only model. This suggests that the joint fusion model benefits from the combination of histopathology and expression data to uncover relevant cancer pathways.

There are a couple of important remarks that come with the proposed methodology. Firstly, the training and feature extraction for the histopathology data is patch-based, meaning that the

model aims at predicting survival scores for each patch. However, in model testing, the risk scores of all the patches of a patient are averaged to get the final risk score. As such, during training we do not leverage multiple patches to obtain the prediction for one WSI. One reason for reducing the number of patches per slide during training is hardware limitations. If multiple patches per slide would be used to obtain predictions during training, the total number of slides per training batch would need to be reduced. However, the Cox loss requires many slides per training batch to learn the differences in survival times. Therefore, given the computational restraints, it was not possible to increase the number of used patches per slide during training. Alternative approaches could be considered for potential improvement. One option is using a different metric other than Cox loss, which does not require many slides per training batch. For example, the survival problem could be structured as a classification problem, an approach that has already been applied for the prediction of lung cancer prognosis[62].

Secondly, although these last few years other and newer CNN architectures have become available, such as DenseNet, EfficientNets, and InceptionNets[63], for the histopathology model a ResNet-50 CNN architecture[39] was chosen. While newer, more complex models might add value they often come with additional computation time, cost, and potential memory explosion. ResNets on the other hand provide a good balance between computation needs and model performance. But, if hardware limitations are not an issue, it could be worthwhile to explore some newer CNN architectures, especially for unimodal model development. As multimodal fusion approaches already have additional computational needs, ResNet architectures provide a good compromise between too complex and too simple models. Furthermore, it has already proven its value as a good model for clinical images, also in brain tumors[64,65].

A third remark concerns the variety of multimodal approaches. The main aim of multimodal learning is to achieve higher accuracy than when using a single modality by integrating mutually complementary information from multiple modalities. So far, most research on multimodal frameworks concerns the development of effective data fusion approaches across different modalities. One important decision is at what specific modeling stage the data fusion will take place. Here, we investigated early fusion, late fusion, and joint fusion strategies. While all three results in good performances compared to unimodal models, in our experiments early fusion seems to be the most robust strategy overall. However, since different modalities cover data in different forms, each modality has specific data distributions, data volume, noise, and thus a different convergence rate. Indeed, learning different modalities at the same rate often leads to (i) a decreased performance, or (ii) a negligible better performance compared to the single data models. Suboptimal learning rates thus prevent maximal use of the potential of combining complementary modalities[66]. In our experiments, for every single modality a grid search was performed to find the optimized learning rate per cohort for the unimodal model. In the early and late fusion approaches, these modality-specific learning rates were kept, and only the learning rate of the final Cox module was optimized. The joint fusion model is the most complex multimodal framework where both modalities are learned simultaneously. To tackle the higher volume of the histopathology data, different learning rates are used, with a higher learning rate for the histopathology data and a smaller learning rate for the genomic data as well as a different learning rate for the last Cox module. These multimodal learning rates were determined via grid search and fine-tuned by empirical testing. But, although the fusion of data modalities boosted the survival predictions in both our training cohorts and external cohorts, the multimodal frameworks might benefit from more optimized learning rates. The idea to find modality-specific

learning rates has recently gained attention and methods are becoming available[66–68]. While these are currently still computational intense, optimized modality-specific learning rates will avoid overfitting of some modalities and underfitting of others, and ultimately lead to (i) better representation of each modality, (ii) detection of the relevant biological interplay between modalities, and (iii) improved generalization to different cohorts.

Lastly, while we show that our multimodal approaches have better performance for prognosis prediction than using each modality in isolation, further evaluation is needed to compare our models with existing clinical models. Stratification in current clinical practice is based on a combination of clinical and/or pathological and/or molecular features. In comparison, our approach aims to handle this heterogeneity in one model. While our results suggest that data fusion is a promising strategy to address the differences between diagnostic subcategories in a unified framework, an additional assessment is needed to proof the value and utility in a real clinical setting.

In summary, for complex diseases such as brain tumors, single data modalities are often not sufficient to disclose the disease trajectory. Neither genomic features nor imaging completely explains the behavior and variable prognosis of brain tumors in pediatric or adult patients. DL methods have now enabled combining these heterogeneous data sources such as images and genomic data. Here, we showed that a combined model is better at capturing patient prognosis. Moreover, interpretation of the joint molecular-imaging model shows that selected genes define molecular pathways that have previously been involved in determining the prognosis of brain tumor patients. This suggests that a multimodal deep learning approach is a promising way of capturing the disease trajectories of brain tumor patients. In future work, the data fusion framework can be expanded to include more data modalities, such as for example radiographic images. CT scans and MRI images have already been used in tumor survival prediction[69] and incorporating them may potentially further improve the survival prediction performance.

## Data availability

Datasets analyzed during the current study were derived from their respective data portals: (i) Adult glioma cohort from The Cancer Genome Atlas (TCGA) available via the GDC data portal (https://portal.gdc.cancer.gov/repository), (ii) Pediatric brain tumor cohort from the Pediatric brain Tumor Atlas (PBTA) available through the Gabriella Miller Kids First Data Resource Portal (KF-DRC, https://kidsfirstdrc.org) and (iii) Adult glioblastoma cohort from the National Cancer Institute's Clinical Proteomic Tumor Analysis Consortium (CPTAC) with RNA-seq data available at the GDC data portal (https://portal.gdc.cancer.gov/repository) and the pathology images at The Cancer Imaging Archive (TCIA) Portal (https://www.cancerimagingarchive.net/datascope/cptac/home/). Source data underlying the graphs in the main figures are available as Supplementary Data 1. All other data are available from the corresponding author upon reasonable request.

## Code availability

The source code of the models used to perform the results and analyses presented in this manuscript are available on GitHub at https://github.com/gevaertlab/MultiModalBrainSurvival and Zenodo at https://doi.org/10.5281/zenodo.7644876.

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

## Acknowledgements

We would like to thank Armaury Sabran for his work during the early stages of this manuscript. We are grateful for his insightful ideas and prototypes of the data fusion concepts. We would also like to express our great appreciation to Marija Pizurica for her support and valuable suggestions in performing the HoverNet experiments. This research was conducted using data made available by The Children's Brain Tumor Tissue Consortium (CBTTC). Research reported here was further supported by the National Institute of Biomedical Imaging and Bioengineering (NIBIB) under award R56 EB020527, and the National Cancer Institute (NCI) under awards: R01 CA260271, U01 CA217851, and U01 CA199241. The content is solely the responsibility of the authors and does not necessarily represent the official views of the National Institutes of Health (NIH).

## Author contributions

S.S. and Y.L.Q. designed and performed the experiments, derived the models, and analyzed the data. S.S. and Y.L.Q. wrote the manuscript with the support of O.G. and Y.Z. Y.Z. aided in the interpretation of the results. P.M. helped with the initial implementation of the model architectures and computational frameworks. H.V. and O.G. contributed to the design and sample selection of the research. O.G. supervised the project. All authors provided critical feedback and helped shape the research, analysis, and manuscript.

## Competing interests

The authors declare no competing interests. Olivier Gevaert is an Editorial Board Member for *Communications Medicine*, but was not involved in the editorial review or peer review, nor in the decision to publish this article.
