## [Peer Review File · Communications Medicine]

Reviewers' comments:

Reviewer #1 (Remarks to the Author):

this is an interesting technical study in which deep learning is used to predict prognosis in glioma. The novelty lies in combination of histology and genetic information, which, although similar to Mobadersany et al PNAS, is broader in scope. The writing is clear and the study protocol makes sense. However, the results of the multi-modality approach only show a minimal increase over single-modality approach so that it is ultimately unclear if the results are clinically meaningful. Also, the authors do not acknowledge a lot of relevant papers which have addressed similar problems, see below.

a few other comments, major and minor

1. Source codes are not available, this is not acceptable. Please put your codes to Github like other papers in this community, it should look like this:

<https://github.com/mahmoodlab/CLAM> or this <https://github.com/mahmoodlab/CRANE>

2. The authors need to disclose adherence to TRIPOD / STARD and include relevant checklists

3. please also discuss some more recent approaches such as this

<https://github.com/mahmoodlab/MCAT> and

<https://www.frontiersin.org/articles/10.3389/fonc.2021.788740/full>. Also, the authors definitely need to cite <https://ieeexplore.ieee.org/document/9186053>

4. include an ethics statement

5. more detail on "add random color jitter" is required

6. please explain in more detail how you tuned hyperparameters

7. Line 188-189, the citation Xu et al. is not really representative of this huge field of patch-based computational pathology, please discuss more broadly. In particular, lots of the work which the authors have done was already presented by Fu et al., Nat Cancer 2020

8. Figure 2 does not look nice and there is much wasted space in this figure

9. include a data availability statement

10. In table 1, the column "Percentage of samples with survival times larger than the median survival time of the entire cohort" is not clear to me, the median is always 50:50. On the other hand, this table lacks a lot of clinically relevant information, such as exact WHO grade, other histological subtypes, tumor site, etc

11. Line 294-295 is speculative and should be supported by experimental data

12. Line 289-291 these hyperparameters are arbitrary and should be tuned on a dedicated

tune set

13. Line 359-360 please provide these results.

14. please compare significance between rows in Table 2.

15. Figure 6 does not look very nice and it needs scale bars

16. Also Figure 7 could be much nicer in terms of visual quality. In general, the figures should have a consistent visual style.

Reviewer #2 (Remarks to the Author):

This is an interesting paper that addresses an exciting area of biomedical research, ie, the application of multi-modal machine learning approaches to generate predictive models of complex diseases. To this end, they developed a unified model for brain tumor survival prediction that uses an integrate multiple modalities including histopathology image data, RNA sequencing data, and clinical data. They show that the integrated model achieved better predictive strength and any single modality alone, although the improvement of the integrated models over the best performing single modality model is modest. They also show that the models could be transferred across tumor types with good performance. While the general approach and findings were clear, many of the details were not clear, making the significance of the specific findings difficult to interpret. Also, because of the lack of details in the methods and results, I think it would be very difficult for someone to reproduce the work and to confirm that they got the same results.

Specific comments:

In the methods section describing the histopathology data analysis they discuss in general terms the extraction of patch-level histopathological features from digitized images of H&E stained tissue sections. It would be helpful if they provided more information of what kinds of features are extracted, and also provide some clear examples of how these features varied across tumor types and how they correlated with survival metrics. This would help readers to understand what type of histopathological features are being extracted, for example, cellularity, nuclear pleomorphism. They do something of this sort in figure 6, but it is very hard to interpret what they are showing. The saliency maps overlaid with the original image could be a nice way to illustrate this point, if it were better explained, and if the histopathological features that they are highlighting were more clearly and definitively identified.

There are similar concerns regarding the genomic model and the very general way in which the data, methods and results are described. For example, in addition to providing the top gene expression pathways, they should also provide the gene lists that were generated from their analysis and give more details on how the pathway analysis was performed.

More information is needed regarding the TCGA data. They included both low grade glioma (LGG) and glioblastoma (GBM), and presumably it is the survival differences between LGG

and GBM that drives the predictive strength of the models. It would be helpful if they explained how the LGG and GBM samples parsed between the low risk and high risk groups. They should also consider the genetic heterogeneity that exists within each of these groups and discuss how such heterogeneity might affect the models. For example, adult low grade gliomas will include astrocytomas and oligodendrogliomas, which have significant differences in survival, histological features, and genomic features. As far as I can tell, the models are blind to these diagnostic categories, but is it fair to assume that the performance of the models would be affected by these associations?

I am not sure what to make of the transfer learning from adult to pediatric glioma and from glioma to ependymoma and medulloblastoma. This suggests that the models are being trained on features that generalize across these very different types of tumors. It would be helpful if the authors could discuss what they see as the significance of these findings. Does this generalizability provide any further insight into the types of features that are driving the performance of the models? Or is the importance of this finding strictly related to the potential to train a model on a larger data set and then apply it to a smaller data set?

One minor comment is that they refer to ependymoma and medulloblastoma as “two more rare pediatric brain tumor types” it is not clear what they mean by “more rare”, since medulloblastoma and ependymoma are 2 of the most common types of pediatric brain tumors. They should provide more precise discussion of the relative frequency of these tumors along with a reference.

Reviewer #3 (Remarks to the Author):

The authors present a multi-modal data fusion model that includes histopathology, gene expression, and clinical data to predict survival in adult (N = 965) and pediatric (N = 305) brain tumor patients. CNN was used for histopathology feature extraction and MLP was used for gene expression data. Clinical data was used in two of data fusion strategies as raw inputs. The authors tested three data fusion strategies: feature fusion, late fusion, and joint training fusion. The authors explain the details of the training procedures very well in the Methods section. They also perform transfer learning experiments across several cohorts.

The models were evaluated using concordance index and Brier score. They were able to achieve CI of 79% and Brier score of 13% in adult patients using Pathology + RNA, and 95% and 8% in pediatric patients using a joint training fusion model.

Strengths

- 1) The manuscript is well written.
- 2) The authors are clear in the introduction and explanation of the topics and methods
- 3) The data is well presented in figures 1, 2, 4, 5.

Weaknesses

- 1) The results are presented are not convincing that the proposed multimodal fusion models provide a clinically significant increase in prognostication over gene expression alone. There

are two reasons for this.

- a) The difference in performance is very small or zero (pediatric gliomas, brier score).
- b) The models are undertested. The authors report on a single iteration of train-validation-test data splitting from an open-source dataset, which is insufficient.

Because the prediction difference is very small, the authors must perform one or both of the following experiments to be convincing.

- 1) Multiple iterations of train-validation-test dataset splitting and report the aggregated results as mean and standard deviation. This will evaluate model stability. These experiments are not computationally prohibitive.
- 2) Test the model on an external dataset, either from their own institution or from another open-source repository.

Lacking these two experiments, the paper does not meet the necessary standards for publication

Reviewer #4 (Remarks to the Author):

I write this review as a translational oncologist, who has expertise in the application of genomics to predicting outcome in paediatric brain tumours and an emerging research interest in the application of deep learning approaches to classify H&E images in combination with omics readouts (transcriptomics, methylomics) for treatment stratification. Having said that, the mathematics shown in lines 204-216 is beyond me and I cannot comment on its appropriateness and hope that other reviewers are able to verify this. I believe this to be a novel approach, however there is not sufficient detail to reproduce the work in its current form.

Qiu and colleagues describe a multi-modal analysis, fusing transcriptome information with histology and clinical data, alongside standard clinical data to predict outcome in a large glioma dataset (paediatric and adult, total $n > 1000$). They also apply the classifiers to medulloblastoma (a neuronal tumour) and ependymoma (a ciliated epithelial glial cell tumour) datasets. They evaluate the performance of the fused classifiers to predict survival in comparison to classifiers that consider clinical data, gene expression and histopathology in isolation. This is an interesting and ambitious paper, however there are some issues that require clarification, and, in particular, some concerns regarding the transfer learning applied to the other brain tumour entities.

The paper is generally easy to follow, but the use of the present tense is unusual and the authors should consider whether this decision is in line with this journal's stylistic rules.

The rationale for the data integration and its comparison to data types in isolation is justified. However, the description of the large cohorts could appear to be misleading, since from the initial description of 450 Low-grade glioma (LGG) and 515 Glioblastoma Multiforme (GBM), there are only 159 samples with available RNA-seq and transcriptome micorarray for 356 samples – total $n = 515$. Does this mean that all LGG lacked transcriptome data or is this a

coincidence? Could you provide a figure that explores issues of combining the RNA-seq and the microarray. It wasn't clear to me which tumour types were microarray and which were RNA-seq. Were there any remaining batch effects or potential confounding?

The creation of the predictors and the resultant fusion classifiers was clearly explained for the most part and the three different fusion classifiers was also clear, however I have a few questions:

- A major issue is that when creating classifiers that use clinical data, there are many other clinical (and molecular) factors that are important for survival prediction and it seems as though the predictive models created here should really be assessed versus current clinical classification, rather than just age and sex. As an example, medulloblastoma outcome prediction is predicated on the assessment of high-risk (i.e. metastatic disease, subtotal resection, patient aged <3 years at diagnosis, large-cell/anaplastic histology, MYC/N amplification, TP53 mutation and SHH subgroup) and favourable risk (CTNNB1 mutant), and any comparisons from the classifiers created here should be assessed in this context. I am less expert in glioma, but there are molecular subgroups of gliomas defined by characteristic mutations.
- I had the impression that the clinical data was used in the fusion classifiers, in addition to the transcriptome and image data, but this data is not shown in the results
- Were the histopath only classifiers trained on the entire dataset, or the subset that also had transcriptome data available?
- When selecting images, how do you account for non-informative tiles – e.g. tiles that span blood vessels, tissue folding, crushing, areas of bleeding, or even tiles without tissue?
- The authors report that there was a 50/50 split for the ependy/medullo datasets (line 178, methods). Did the authors train models based on these small datasets? If so, I couldn't see where they were reported. I suppose that these models didn't perform well but if mentioned in the methods, they should be shown.
- For tumours with >1 slide available, does that potentially bias their contribution to the predictor, since more tiles/patches will be considered for these patients in comparison with patients with only 1 slide available?
- For the flash-frozen tissue sections, in my experience, they provide less resolution and detail than FFPE-sections, and it might be more informative to analyse the FFPE samples. Do you think this could be problematic?
- Transfer learning - given that the other tumour cohorts are already small (47 ependy, 60 medullo), it is difficult to justify the 50:50 split into train/test prior to model validation and are arguably too small to assess prognostic capability.
- How were the train/test sets applied in this study? I didn't see that the ependy and medullo predictors were assessed?
- If the authors think that the application of the transfer learning to the ependy and medullo sets is justified, the Kaplan-Meier curves should be shown, in addition to the Cox model outputs so that the significance of the predictor can be assessed.
- While ependymoma is a glial tumour, it is difficult to see the rationale for the transfer learning applied to medulloblastoma, a neuronal tumour that has its own distinct clinico-molecular disease behaviour that is very different to gliomas and this decision should be justified in more detail.

The first results deal with the creation of classifiers from the adult and paediatric glioma cohorts. For adult glioma, remarkably the very simple age/sex classifier performs equivalently to the more complex predictors created – if an age/sex classifier performs comparably, what is the advantage of invoking these more complex predictors? However, for the paediatric glioma sets, the RNA-data alone and one of the fusion classifiers perform equivalently and are far superior to the clinical model. Why do the Kaplan-Meier curves in figures 4 and 5 not include the curves for the late fusion model?

For the paediatric glioma sets, the predictors pick out a group of patients with favourable risk and a 100% survival – it would be good to see at-risk tables to see the differences between the number of patients assigned to the favourable risk group in each panel, since the favourable-risk curves look similar.

The transfer learning between the adult and paediatric glioma datasets should also be accompanied by Kaplan-Meier plots (at least in the supplementary data), along with the accompanying Cox models, to enable the reader to assess the models directly. The Brier scores and Concordance indices are noticeably lower, so perhaps the models do not transfer particularly well.

In the latter part of the paper, the authors back-propagate the predicted risk to identify features important for outcome prediction. Could you provide details on how this was done in the methods, I didn't see where it was mentioned?

The histology data shown in Figure 6 are interesting, but the indicated areas in figure 6a (top left of the right hand panel), appear to be non-tumour (perhaps neuropil, though resolution is low). Is this problematic? Which tumour type is figure 6? Could the survival be ascribed to the upper panel (a) having larger nuclei compared to panel (b)? Large nuclei are typically associated with poorer-survival.

In figure 7, the same approach is applied to the transcriptome data, and specific reactome pathways with non-random gradient scores are shown. Is this result significant? How do the pathways compare versus genes chosen at random? It would be good to see some significance scores if possible to assess whether these are meaningful. Have the most important genes previously been associated with survival in glioma and/or the other tumour types?

The brief discussion recapitulates what was done, but I was confused with the discussion regarding the number of patches – in the methods, lines 374-7, it states that a number of different patches were used to train the model, but that you settled on 100 patches. In the discussion, lines 569-572, the authors state that there is a limit of 2-3 patches per slide in practice. Could you clarify what you mean here? The discussion is too short, and should consider the study limitations in more depth, and also consider whether this approach could augment existing clinical markers.

Reviewer #5 (Remarks to the Author):

In their manuscript entitled “Multi-modal data fusion of adult and pediatric brain tumors with deep learning”, Qiu and colleagues developed a multimodal deep learning model and trained it on a combination of histopathological and RNA expression data to predict the prognosis of patients with pediatric and adult brain tumors. To this end they developed three different strategies for data fusion and training of the network. They looked at different pediatric as well as adult brain tumor types and used one data set for each of these tasks. As performance metrics, Harrel’s concordance index (C-Index) as well as the integrated Brier score (IBS) were used. Furthermore (as far as I understand), C-Index was used to divide the cohort into low- and high-risk patients for survival analyses. Lastly, image as well as RNA expression features were extracted and described. While the underlying idea -combining various data sources for more precise prognosis prediction- is innovative and promising, there are various points of concern:

1) Novelty

In my opinion, the study by Qiu et al. does neither substantially advance our knowledge about the use of multimodal deep learning models nor does it pose a significant improvement over previous studies. Similar approaches have been used for both brain cancer [1, 2] as well as other types of diseases [3-6]. What do the authors regard as substantially different to previous approaches?

2) Methodology and Interpretation (Major)

a) The biggest problem is that no independent external test set was used. It is completely unclear whether the fusion model is able to generalize and perform well on new data from different cohorts. We know that deep learning models are very good in “learning” (or better “memorizing”) cohort specific data structures, but then fail on similar data from external sources. This holds true for fusion models in particular as they tend to get rather “deep”. The necessity of using an independent external cohort is highlighted i. e. by Kleppe et al. and the authors should consider following the proposed PIECES recommendations and other AI guidelines [7].

Furthermore, a strong indication that the models are indeed overfitting the training data can be gathered from the fact that there is a dramatic drop in performance when models trained on pediatric tumors are used on adult tumors and vice versa.

b) It appears that the authors did not even use any type of k-fold cross validation either. They provide C-Index and IBS values for each modality together with some sort of confidence interval, but it is not clear where this confidence interval even comes from. It seems somewhat odd that the confidence interval is ± 0.02 for all experiments on adult glioma (Table 2). Where do these values originate from and how do the authors explain this? Furthermore, there is no statement on whether the differences in performance (between the single modality models and the fusion models) are even statistically significant. To me the “improvement” by the combined approach seems only marginal and falls well within the margin of error (i. e. Table 1: RNA data alone: 0.78 ± 0.02 vs. Pathology + RNA: 0.79 ± 0.02). This holds true for each model presented throughout the entire manuscript. Just using a bold font does not make a model significantly better.

c) C-Index as well as IBS can only be used to evaluate the performance on an entire cohort of patients. For the purpose of clinical translation this is problematic as you cannot make predictions on an individual patient. This also limits the meaningfulness of the data presented with the Kaplan-Meier-Curves (Figures 4 and 5) as the presented cut-off would be highly specific only for patients within the cohort.

d) While the authors should be commended for including “explainability” experiments, the

results presented in Figure 6 are not convincing. The image quality is poor, there are no scale bars, and neither an overview of the WSI nor a magnification of any cellular details are provided. Thus, the image features mentioned in the text cannot be recognized. The saliency maps do not point to a single feature (as would be expected) but are scattered throughout the entire image. The sarcomatoid morphology of the tumor shown in the figure cannot be recognized and is not highlighted by the heatmap. If this was the case, I would expect a much stronger signal colocalizing with the most pleomorphic cells and nuclei. If the other cells highlighted truly represent apoptotic cells, why exactly would the authors expect a poorer survival? Wouldn't an increase in apoptosis of the tumor be beneficial for the patient? A systematic analysis of the detected image features by an expert in (neuro)pathology is missing.

3) Methodology and Interpretation (Minor):

a) It is unclear what type of ResNet was used and if the authors have tried other / newer architectures (DenseNets, EfficientNets, ViTs, etc.)? Furthermore, I don't really understand the reasoning behind not using any type of color normalization? Only using augmentation could be limiting performance.

b) Have the authors tried including WSIs from FFPE samples (sometimes referred to as diagnostic slides) of the TCGA cohort? Tissue morphology can present totally different in frozen sections as opposed to FFPE which again can influence a model's performance.

c) This might just be me, but I find it a bit of a stretch to refer to "age" and "gender" as "clinical data" in this context. I would usually use the term "clinical data" for other parameters such as blood pressure or EEG measurements. It would be much more interesting if the authors actually included other types of data such as radiologic images or laboratory parameters.

d) The link to the provided Github repo does not work and I could not access the underlying code base.

1. Chen, R.J., et al., Pathomic Fusion: An Integrated Framework for Fusing Histopathology and Genomic Features for Cancer Diagnosis and Prognosis. *IEEE Trans Med Imaging*, 2020. PP.
2. Mobadersany, P., et al., Predicting cancer outcomes from histology and genomics using convolutional networks. *Proc Natl Acad Sci U S A*, 2018. 115(13): p. E2970-E2979.
3. Liu, Y., et al., A deep learning system for differential diagnosis of skin diseases. *Nat Med*, 2020. 26(6): p. 900-908.
4. Ning, Z., et al., Integrative analysis of cross-modal features for the prognosis prediction of clear cell renal cell carcinoma. *Bioinformatics*, 2020. 36(9): p. 2888-2895.
5. Schulz, S., et al., Multimodal Deep Learning for Prognosis Prediction in Renal Cancer. *Front Oncol*, 2021. 11: p. 788740.
6. Huang, S.C., et al., Fusion of medical imaging and electronic health records using deep learning: a systematic review and implementation guidelines. *NPJ Digit Med*, 2020. 3: p. 136.
7. Kleppe, A., et al., Designing deep learning studies in cancer diagnostics. *Nat Rev Cancer*, 2021. 21(3): p. 199-211.

Response to reviewers

We thank the reviewers for their thorough analyses of our manuscript. We have reworked the manuscript and the figures extensively and believe that the manuscript has now been significantly improved.

Reviewer #1:

This is an interesting technical study in which deep learning is used to predict prognosis in glioma. The novelty lies in combination of histology and genetic information, which, although similar to Mobadersany et al PNAS, is broader in scope. The writing is clear and the study protocol makes sense. However, the results of the multi-modality approach only show a minimal increase over single-modality approach so that it is ultimately unclear if the results are clinically meaningful. Also, the authors do not acknowledge a lot of relevant papers which have addressed similar problems, see below.

1. Source codes are not available, this is not acceptable. Please put your codes to Github like other papers in this community, it should look like this: <https://github.com/mahmoodlab/CLAM> or this <https://github.com/mahmoodlab/CRANE>

We indeed added the wrong link to the GitHub repository. We have now added a “Data Availability” section (Lines 680-690) including the correct location of the GitHub repo containing the source code of the models.

2. The authors need to disclose adherence to TRIPOD / STARD and include relevant checklists

We completely agree and understand the importance of adhering to these checklists. In this revision we have followed the journal guidelines in completing the TRIPOD and STARD lists.

3. please also discuss some more recent approaches such as this

<https://github.com/mahmoodlab/MCAT> and <https://www.frontiersin.org/articles/10.3389/fonc.2021.788740/full>

Also, the authors definitely need to cite

<https://ieeexplore.ieee.org/document/9186053>

We fully concur with the reviewer and acknowledge that the introduction lacked a description of other, recent approaches. As the reviewer suggests, we have now added a paragraph (Lines 93-120) with an overview outlining some recent multimodal approaches with similar goals.

4. include an ethics statement

We have now included an “ethics statement” at the bottom of the manuscript (Lines 691-702).

5. more detail on "add random color jitter" is required

We agree with the reviewer to add more explanation about the addition of color jitter as an augmentation step. Lines 203-214 contain more details about the preprocessing of the histopathology images including the stain augmentation step. When working with H&E slides some kind of normalization or additional augmentation is essential for performance to account for variability and H&E stains. As in [1] the authors show that augmentation is the most important for CNNs and that additional normalization adds limited value on top; In our work, we opted for a stain augmentation by adding random color jitter using Pytorch’s ColorJitter function to mask the H&E stains.

[1] Tellez, D., et al., Quantifying the effects of data augmentation and stain color normalization in convolutional neural networks for computational pathology. *Med Image Anal*, 2019. 58: p. 101544. 10.1016/j.media.2019.101544

6. please explain in more detail how you tuned hyperparameters

We split the data in a test set and training set at a 80/20 ratio. On the training set we performed a cross validation approach (10-fold for adult cohort, 5-fold for pediatric cohort) and performed a grid search to tune the hyperparameters. The optimal weights for each fold (CV configuration) were saved based on the epoch that achieved the highest validation accuracy. The best model was chosen based on the fold that gave the lowest validation loss and was evaluated on the test set. As suggested, we added more details about model training and tuning to the manuscript (Lines 312-370).

7. Line 188-189, the citation Xu et al. is not really representative of this huge field of patch-based computational pathology, please discuss more broadly. In particular, lots of the work which the authors have done was already presented by Fu et al., Nat Cancer 2020

While the cited paper of Xu et al [1] introduces the adoption of patch sampling and pooling techniques to leverage whole slide histopathology images while making the whole framework scalable and efficient on extremely large, their specific approach indeed differs from our work. The approach described in Fu et al [2] as well as Murchan et al [3]) are indeed more in line with our used methodology. As such, we have updated the text of the specific section and added these citations (Lines 202-203).

[1] Xu, Y., et al. Large-scale tissue histopathology image classification, segmentation, and

693 visualization via deep convolutional activation features. BMC bioinformatics 2017;18(1):1-17.

[2] Fu, Y., et al., Pan-cancer computational histopathology reveals mutations, tumor composition and prognosis. Nat Cancer, 2020. 1(8): p. 800-810. 10.1038/s43018-020-0085-8

[3] Murchan, P., et al., Deep Learning of Histopathological Features for the Prediction of Tumour Molecular Genetics. Diagnostics (Basel), 2021. 11(8). 10.3390/diagnostics11081406 [3] Murchan et al. (2021, Diagnostics)

8. Figure 2 does not look nice and there is much wasted space in this figure

This is a very valid comment. We have reworked this Figure (now Figure 3) and also the other figures to make it visually more appealing. We hope this updated figure will make it easier to grasp the concepts described in the main text.

9. include a data availability statement

We have now included a data availability statement (Lines 680-690) containing the link to the GitHub repo with the source code as well as information on the used datasets.

10. In table 1, the column "Percentage of samples with survival times larger than the median survival time of the entire cohort" is not clear to me, the median is always 50:50. On the other hand, this table lacks a lot of clinically relevant information, such as exact WHO grade, other histological subtypes, tumor site, etc

Former Table 1 was indeed confusing and didn't add a lot of value to the manuscript. In addition, as opposed the approach in the previous manuscript, in this revision we use a cross-validation strategy. For these reasons, we discarded the table in this revision and choose to explain in the methods section (Lines 313-322) how we defined and generated the training, validation, and test sets.

11. Line 294-295 is speculative and should be supported by experimental data

Without displaying any supportive data or experiment, we agree that the claim in lines 294-295 is more hypothetical. However, while revising the paper, we added new analyses and did a comprehensive re-writing. In addition, we switched the focus solely to the histopathological and genomic modalities and their complementary potential. Therefore, the claim in lines 294-295 are no longer of relevant.

12. Line 289-291 these hyperparameters are arbitrary and should be tuned on a dedicated tune set

We fully agree that these (and all learning rates) should be tuned on a separate validation set. In our experiments, a cross validation grid search was performed for each single and multimodal model to find the optimized learning rate per cohort. However, the multimodal frameworks might benefit from more complex approaches to find the most optimal learning rates. In the discussion we now dedicated a separate section where we elaborate more on this topic (Lines 655-662).

13. Line 359-360 please provide these results.

A statement as the one made in line 359-360 indeed needs supporting evidence. As already mentioned above, we made some major updates in our analyses. One update is that we have added a cross-validation strategy to develop and tune the models. In these analyses, we immediately started from ResNet-50 that was pretrained on the ImageNet dataset. Using the weights of a CNN model pretrained on a large dataset, such as ImageNet, has some advantages over learning the weights from scratch: reducing the need for a large dataset, as well as reducing training time and computational resources. Especially for clinical applications this is relevant, as a large, labeled dataset is often not available, and usage of ResNet models pretrained on the non-medical ImageNet has already shown promising results for medical data analysis [1]. While using pretrained models of the ImageNet dataset is still one of the most popular methods in medical data analysis, it is not always a given that it offers significant improvements in all medical imaging domains [2]. However, in our current work, we did not perform a comparison between a pretrained vs non-pretrained ResNet50. But, while out of scope for this manuscript, this comparison (and also comparison between different architectures) could be of interest for future follow-up work.

[1] Morid MA, Borjali A, Del Fiol G. A scoping review of transfer learning research on medical image analysis using ImageNet. *Comput Biol Med.* 2021 Jan;128:104115. doi: 10.1016/j.compbimed.2020.104115. Epub 2020 Nov 13. PMID: 33227578.

[2] Alzubaidi L, Duan Y, Al-Dujaili A, Ibraheem IK, Alkenani AH, Santamaría J, Fadhel MA, Al-Shamma O, Zhang J. Deepening into the suitability of using pre-trained models of ImageNet against a lightweight convolutional neural network in medical imaging: an experimental study. *PeerJ Comput Sci.* 2021 Sep 28;7:e715. doi: 10.7717/peerj-cs.715. PMID: 34722871; PMCID: PMC8530098.

14. please compare significance between rows in Table 2.

We agree that instead of solely reporting the results for each model, a significance test was lacking. After performing cross validation, we performed a pairwise comparison between each strategy (Wilcoxon rank sum) to evaluate the scores. This was done for both the validation sets

as well as the test set. Scores are also now visualized in box plots (Figure 5 and Supplementary Figure 2) together with indication of the significant differences between strategies.

15. Figure 6 does not look very nice and it needs scale bars

Accommodating comments from other reviewers about the lack of biological significance of these type of visualizations, we removed this figure from the manuscript.

16. Also Figure 7 could be much nicer in terms of visual quality. In general, the figures should have a consistent visual style.

We completely agree that the figures were not consistent and lacked quality. We have reworked all the figures to make them more visually appealing with a unified style and color scheme amongst them, less lost space, and overall, more intuitive visualizations.

Reviewer #2:

This is an interesting paper that addresses an exciting area of biomedical research, ie, the application of multi-modal machine learning approaches to generate predictive models of complex diseases. To this end, they developed a unified model for brain tumor survival prediction that uses an integrate multiple modalities including histopathology image data, RNA sequencing data, and clinical data. They show that the integrated model achieved better predictive strength and any single modality alone, although the improvement of the integrated models over the best performing single modality model is modest. They also show that the models could be transferred across tumor types with good performance. While the general approach and findings were clear, many of the details were not clear, making the significance of the specific findings difficult to interpret. Also, because of the lack of details in the methods and results, I think it would be very difficult for someone to reproduce the work and to confirm that they got the same results.

Specific comments:

1. In the methods section describing the histopathology data analysis they discuss in general terms the extraction of patch-level histopathological features from digitized images of H&E stained tissue sections. It would be helpful if they provided more information of what kinds of features are extracted, and also provide some clear examples of how these features varied across tumor types and how they correlated with survival metrics. This would help readers to understand what type of histopathological features are being extracted, for example, cellularity, nuclear pleomorphism. They do something of this sort in figure 6, but it is very hard to interpret what they are showing. The saliency maps overlaid with the original image could be a nice way to illustrate this point, if it were better explained, and if the histopathological features that they are highlighting were more clearly and definitively identified.

The reviewer is right that the extracted features and the previous Figure 6 might need some additional clarification. Here, we implemented a ResNet-50 model that was pretrained on the ImageNet dataset [1]. A ResNet (Residual Neural Network) is a Convolutional Neural Network that stacks “residual blocks” (i.e. neural network layers) on top of each other ultimately forming a network. ResNets comes in many variations that build on this same definition but with different number of layers. For example, ResNet-50 refers to a ResNet architecture with 50 neural network layers. Here, each layer increases the complexity and enriches the features learned by the model. The first layers will learn more generic image features such as edges, corners, while the last layers learn features more specific to the task of interest such as recognizing different shapes and

objects [2]. ResNet-50 is one of most popular deep learning networks and can be trained/finetuned for many computer vision tasks. Also for digital pathology, they have achieved accuracies higher than classical ML models [3]. However, they remain “black-boxes” that do not directly reveal the features that they use to make predictions. Luckily, methods now exist that try to “backpropagate” the prediction to decompose the network into features. This is what we tried to visualize in Figure 6 of the previous submission. The biological interpretation of these features however is still a challenge. Most methods that try to interpret the neural network reveal “where” a model is looking, but not “why” or “what” information in that region is important [4]. The features that ResNets uses to make predictions are often hard to interpret and need to be distinguished from biological features. For example, in Figure 6, we showed the regions or pixels that the model uses to make prediction from the FF slides. These features are not the same as the usual hand-crafted features, like shape of cells, but consist of abstract patterns and concepts from raw pixels. Note that in this revised work, we used FFPE instead of FF as input of the histopathology models as FFPE better preserves cellular and architectural morphology. However, also accommodating comments raised by other reviewers about the biological significance (or lack thereof) of these visualizations, we decided to not include them in this revision.

[1] J. Deng, W. Dong, R. Socher, L. -J. Li, Kai Li and Li Fei-Fei, "ImageNet: A large-scale hierarchical image database," 2009 IEEE Conference on Computer Vision and Pattern Recognition, 2009, pp. 248-255, doi: 10.1109/CVPR.2009.5206848.

[2] He, K., et al. Deep Residual Learning for Image Recognition. 2015. arXiv:1512.03385.

[3] Noorbakhsh, J. et al. Deep learning-based cross-classifications reveal conserved spatial behaviors within tumor histological images. *Nat. Commun.* 11, 1–14 (2020)

[4] Foroughi pour, A., White, B.S., Park, J. et al. Deep learning features encode interpretable morphologies within histological images. *Sci Rep* 12, 9428 (2022).

<https://doi.org/10.1038/s41598-022-13541-2>

2. There are similar concerns regarding the genomic model and the very general way in which the data, methods and results are described. For example, in addition to providing the top gene expression pathways, they should also provide the gene lists that were generated from their analysis and give more details on how the pathway analysis was performed.

We have now added a more detailed description of the methods and results of the interpretation of the genomic model (Lines 421-432 and Lines 518-543) and go more into depth in the discussion (Lines 594-610). It is important to note that the performed pathway analysis differs from a “standard” pathway analysis in a couple of ways. While a standard pathway analysis is a kind of enrichment analysis of genes that are different between groups coupled to statistical testing, the foundation of our method are the pathway gradients related to the sample prediction. In other words, which pathways are important for a neural network when making a prediction for a particular data point. Fetching the gradients is a technique used to try to explain how each input

feature affects the model output with the aim to increase transparency and interpretability of machine learning models. Pathway gradients were obtained by first performing a backpropagation of the prediction to get the gene gradients and next mapping the genes to the pathways of the Reactome pathway collection [6]. Importantly, these gene gradients differ from gene expression values and represent the contribution in making the actual prediction. For each pathway, a gradient was obtained by averaging the gradients of its gene set. This was done for all samples and visualized in a SHAP summary plot [7] making it possible to assess what pathways were the most important (or contributed the most) in making the predictions.

[1] Pozo, K. and J.A. Bibb, The Emerging Role of Cdk5 in Cancer. *Trends Cancer*, 2016. 2(10): p. 606-618. [10.1016/j.trecan.2016.09.001](https://doi.org/10.1016/j.trecan.2016.09.001)

[2] Ito, K., et al., PTK6 regulates growth and survival of endocrine therapy-resistant ER+ breast cancer cells. *NPJ Breast Cancer*, 2017. 3: p. 45. [10.1038/s41523-017-0047-1](https://doi.org/10.1038/s41523-017-0047-1)

[3] Jin, W., Roles of TrkC Signaling in the Regulation of Tumorigenicity and Metastasis of Cancer. *Cancers (Basel)*, 2020. 12(1). [10.3390/cancers12010147](https://doi.org/10.3390/cancers12010147)

[4] Vladimirova, V., et al., Runx2 is expressed in human glioma cells and mediates the expression of galectin-3. *J Neurosci Res*, 2008. 86(11): p. 2450-61. [10.1002/jnr.21686](https://doi.org/10.1002/jnr.21686)

[5] Sun, J., et al., RUNX3 inhibits glioma survival and invasion via suppression of the beta-catenin/TCF-4 signaling pathway. *J Neurooncol*, 2018. 140(1): p. 15-26. [10.1007/s11060-018-2927-0](https://doi.org/10.1007/s11060-018-2927-0)

[6] Gillespie, M., et al., The reactome pathway knowledgebase 2022. *Nucleic Acids Res*, 2022. 50(D1): p. D687-D692. [10.1093/nar/gkab1028](https://doi.org/10.1093/nar/gkab1028)

[7] Lundberg, S.M. and S.-I. Lee, A Unified Approach to Interpreting Model Predictions, I. Guyon, et al., Editors. 2017.

3. More information is needed regarding the TCGA data. They included both low grade glioma (LGG) and glioblastoma (GBM), and presumably it is the survival differences between LGG and GBM that drives the predictive strength of the models. It would be helpful if they explained how the LGG and GBM samples parsed between the low risk and high-risk groups. They should also consider the genetic heterogeneity that exists within each of these groups and discuss how such heterogeneity might affect the models. For example, adult low grade gliomas will include astrocytomas and oligodendrogliomas, which have significant differences in survival, histological features, and genomic features. As far as I can tell, the models are blind to these diagnostic categories, but is it fair to assume that the performance of the models would be affected by these associations?

Yes, you are exactly right, gliomas are very heterogeneous in both morphological and genomic features with variable prognosis. While tumor subtyping and grading based on histology aids to guide treatment trajectory, this comes with the major limitation that differences in subtypes can be very subtle and that there can be a lot of variation in decision making between observers. Multiple studies now have shown that molecular subtyping (based on tumor expression profiles)

are actually a more accurate and objective method to classify tumors and correlates better with prognosis [1]. To tackle the variability amongst gliomas, we aimed to train one model that could handle this heterogeneity. Therefore, we implemented a comprehensive approach that by fusing both molecular and morphological features leverages information from different scales to make better survival predictions. We validated the robustness of this approach by (i) applying the adult framework on an independent more homogeneous glioma cohort and (ii) transferring the pediatric models to two different pediatric subtypes. In both cases, the data fusion model showed better performance for prognosis prediction than using each modality in isolation. This suggests that data fusion is a promising strategy to address the differences between diagnostic categories in one unified model.

[1] Gravendeel, L.A.M., et al., Intrinsic Gene Expression Profiles of Gliomas Are a Better Predictor of Survival than Histology. *Cancer Research*, 2009. 69(23): p. 9065-9072.

4. I am not sure what to make of the transfer learning from adult to pediatric glioma and from glioma to ependymoma and medulloblastoma. This suggests that the models are being trained on features that generalize across these very different types of tumors. It would be helpful if the authors could discuss what they see as the significance of these findings. Does this generalizability provide any further insight into the types of features that are driving the performance of the models? Or is the importance of this finding strictly related to the potential to train a model on a larger data set and then apply it to a smaller data set?

This is a great question, and we are happy to elaborate more on this. The importance of these findings is a bit of both: the models find features that are characteristic to specific tumor types, but also identifies features that are cancer(brain-tumor)-specific that can be used for other datasets. This is one of the great properties of deep learning and why it has revolutionized cancer classification and prognosis. Deep learning models consist of multiple layers and each layer contains new abstract information (features) of the data. While many of those are specific to individual cancer (sub) types, others are preserved and generalizable among multiple cancers, often from related tissues [1]. For example, although ependymoma and medulloblastoma are different subtypes each with their own characteristics and different origin, knowledge and features learned on the pediatric glioma dataset might also be of interest for these subtypes dataset. In our transfer learning experiments, we were interested in investigating how the features learned on one cohort, both for single and multimodal models, would perform when used on different brain tumor cohorts. Mostly, this technique is used for datasets with a small sample size making it hard to develop a model from scratch. The effectiveness of transfer learning demonstrated in our framework is potentially helpful for survival predictions in a more general context, i.e. where there is heterogeneity in the patient cohorts, as well as in the context of (pediatric) brain tumor subtypes with small cohort sizes where it is intractable to develop de novo models.

[1] Foroughi pour, A., White, B.S., Park, J. et al. Deep learning features encode interpretable morphologies within histological images. *Sci Rep* 12, 9428 (2022). <https://doi.org/10.1038/s41598-022-13541-2>

5. One minor comment is that they refer to ependymoma and medulloblastoma as “two more rare pediatric brain tumor types” it is not clear what they mean by “more rare”, since medulloblastoma and ependymoma are 2 of the most common types of pediatric brain tumors. They should provide more precise discussion of the relative frequency of these tumors along with a reference.

We agree with the reviewer that the term “more rare” is confusing in this context. By “more rare”, we were referring to the fact that these cohorts have less samples with both histopathology and expression data available. And thus not that these two pediatric brain tumors occur less. We have now updated the manuscript text and clearly specified that we meant this.

Reviewer #3:

The authors present a multi-modal data fusion model that includes histopathology, gene expression, and clinical data to predict survival in adult (N = 965) and pediatric (N = 305) brain tumor patients. CNN was used for histopathology feature extraction and MLP was used for gene expression data. Clinical data was used in two of data fusion strategies as raw inputs. The authors tested three data fusion strategies: feature fusion, late fusion, and joint training fusion. The authors explain the details of the training procedures very well in the Methods section. They also perform transfer learning experiments across several cohorts.

The models were evaluated using concordance index and Brier score. They were able to achieve CI of 79% and Brier score of 13% in adult patients using Pathology + RNA, and 95% and 8% in pediatric patients using a joint training fusion model.

Strengths

- 1) The manuscript is well written.
- 2) The authors are clear in the introduction and explanation of the topics and methods
- 3) The data is well presented in figures 1, 2, 4, 5.

Weaknesses

- 1) The results are presented are not convincing that the proposed multimodal fusion models provide a clinically significant increase in prognostication over gene expression alone. There are two reasons for this.
 - a) The difference in performance is very small or zero (pediatric gliomas, brier score).
 - b) The models are undertested. The authors report on a single iteration of train-validation-test data splitting from an open-source dataset, which is insufficient.

Because the prediction difference is very small, the authors must perform one or both of the following experiments to be convincing.

- 1) Multiple iterations of train-validation-test dataset splitting and report the aggregated results as mean and standard deviation. This will evaluate model stability. These experiments are not computationally prohibitive.
- 2) Test the model on an external dataset, either from their own institution or from another open-source repository.

Lacking these two experiments, the paper does not meet the necessary standards for publication

We fully agree with the reviewer that these two points are crucial to evaluate model performance and that without them the paper is lacking a solid foundation. This is true for all ML/DL experiments for that matter. To accommodate these two comments, we performed new experiments and added the following adjustments:

- FFPE instead of FF:
 - Instead of Fresh Frozen (FF) samples, we retrained the histopathology model on images from Formalin-Fixed Paraffin-Embedded (FFPE) samples. Compared to FF, FFPE better preserves cellular and architectural morphology. FF on the other hand, is more ideal for gene expression analysis (better RNA quality) than FFPE. Therefore, fusing FF with expression data might not add a lot of additional value, explaining why the multimodal models don't show a significant improvement in performance. As such, because FFPE covers a different scale, we hypothesized that the morphological features from FFPE data better complement corresponding expression data and can provide more new information when fusing these two different modalities. As described in the result section and summarized in Table 1 and Figure 5, data fusion of FFPE data with the expression data indeed lead to significant increased performance versus each single modality alone.

- Cross validation
 - As correctly pointed out by the reviewer; multiple iterations of data splitting are needed to make a correct assessment of model performance and stability. In this revised work, we implemented a cross-validation strategy (10-fold for the adult cohort and 5-fold for the pediatric cohort). Models were evaluated on each validation set of the fold as well as on a separate test set that was not used for training (see methods section "Model Training" for more details (Lines 313-322)).

- Updated reporting on model performances
 - As described in the Methods section "Performance Metrics" (Lines 372-399), we defined a new scoring feature that combines the Integrated Brier Score (IBS) and Concordance-Index (CI) into one measure. This "Composite Score" (CS) thus represents an average of a relative measure (CI) and an absolute error metric (IBS) and allows straightforward evaluation and comparison of different models.
 - Model scores are now reported as mean and standard deviation (for CS, IBS, and CI scores) for the cross-validation experiments. In addition, we also performed a pairwise comparison (Wilcoxon rank sum) of the score distributions to evaluate the significance of differences in model performances.

- Validation on an external cohort
 - As suggested by the reviewer, we additionally tested our adult glioma models on a third external validation set, CPTAC, an open-source dataset which holds FFPE and expression data of 97 glioblastoma (GBM) patients. Lines 484-496 and Table 2 show that also on this cohort the multimodal models perform better, with the early fusion again the best model.

We hope that this revised work addresses the issues raised by the reviewer. Overall, we feel that the manuscript greatly benefits from these additions and updates.

Reviewer #4:

I write this review as a translational oncologist, who has expertise in the application of genomics to predicting outcome in paediatric brain tumours and an emerging research interest in the application of deep learning approaches to classify H&E images in combination with omics readouts (transcriptomics, methylomics) for treatment stratification. Having said that, the mathematics shown in lines 204-216 is beyond me and I cannot comment on its appropriateness and hope that other reviewers are able to verify this.

We acknowledge that the section going into depth about the mathematical formulas might have been too technical and could have used more background information for context. Therefore, we have rewritten the introduction and added an introductory section about the Cox model, its definition, how it differs from other models, its benefits and why it's useful for survival predictions. In the following sections we next describe how it can be applied for deep learning applications and list some examples of other work in this field. Alongside the introduction, we also rewrote the methods section "Survival Prediction" where we outline in more detail the specifics of the Cox-Proportional Hazard model and how we apply the formulas in our deep learning frameworks. By reworking these sections, we hope these parts are more comprehensible now and better describe the concepts of our work.

I believe this to be a novel approach, however there is not sufficient detail to reproduce the work in its current form. Qiu and colleagues describe a multi-modal analysis, fusing transcriptome information with histology and clinical data, alongside standard clinical data to predict outcome in a large glioma dataset (paediatric and adult, total $n > 1000$). They also apply the classifiers to medulloblastoma (a neuronal tumour) and ependymoma (a ciliated epithelial glial cell tumour) datasets. They evaluate the performance of the fused classifiers to predict survival in comparison to classifiers that consider clinical data, gene expression and histopathology in isolation. This is an interesting and ambitious paper, however there are some issues that require clarification, and, in particular, some concerns regarding the transfer learning applied to the other brain tumour entities.

The paper is generally easy to follow, but the use of the present tense is unusual and the authors should consider whether this decision is in line with this journal's stylistic rules.

This is a valid comment and we have now rewritten the manuscript in the past tense. We thank the reviewer for pointing this out and we feel this change improved the flow of the manuscript.

The rationale for the data integration and its comparison to data types in isolation is justified. However, the description of the large cohorts could appear to be misleading, since from the initial description of 450 Low-grade glioma (LGG) and 515 Glioblastoma Multiforme (GBM), there are only 159 samples with available RNA-seq and transcriptome micorarray for 356 samples – total n=515. Does this mean that all LGG lacked transcriptome data or is this a coincidence? Could you provide a figure that explores issues of combining the RNA-seq and the microarray. It wasn't clear to me which tumour types were microarray and which were RNA-seq. Were there any remaining batch effects or potential confounding?

We agree that the section outlining the used cohorts and describing the exact samples and data formats was a bit too short and for that reason not super clear. We have now rewritten this section (see Methods section "Data", Lines 142-180) and also added a figure (Figure 1) that provides a thorough overview of the used data and numbers. To specifically answer the reviewer's question about the adult cohort: all the LGG samples (N=426) had both histopathology data and RNAseq data. However, for the GBM samples (N=357), 158 samples had RNAseq data and 199 samples had corresponding microarray data. After preprocessing and standardizing both the RNAseq and microarray data, the Combat-Seq package was used to account for any batch effects between these two platforms for this cohort. Figure 1 now also illustrates this for the adult cohort, but also for the other used datasets. Note that because in this revised work we used FFPE data instead of FF, the total number of available samples is slightly different from the previous submission.

The creation of the predictors and the resultant fusion classifiers was clearly explained for the most part and the three different fusion classifiers was also clear, however I have a few questions:

1. A major issue is that when creating classifiers that use clinical data, there are many other clinical (and molecular) factors that are important for survival prediction and it seems as though the predictive models created here should really be assessed versus current clinical classification, rather than just age and sex. As an example, medulloblastoma outcome prediction is predicated on the assessment of high-risk (i.e. metastatic disease, subtotal resection, patient aged <3 years at diagnosis, large-cell/anaplastic histology, MYC/N amplification, TP53 mutation and SHH subgroup) and favourable risk (CTNNB1 mutant), and any comparisons from the classifiers created here should be assessed in this context. I am less expert in glioma, but there are molecular subgroups of gliomas defined by characteristic mutations.

Yes, you are exactly right, brain tumors are very heterogeneous in both morphological

and genomic features with variable prognosis. While tumor subtyping and grading based on histology aids to guide treatment trajectory, differences in subtypes can be very subtle and multiple studies now have shown that molecular subtyping is actually a more accurate and objective method to classify tumors and correlates better with prognosis [1]. To tackle this variability, we aimed to train one model that could handle this heterogeneity. Therefore, we implemented a comprehensive approach that by fusing both molecular and morphological features leverages information from different scales to make better survival predictions. We validated the robustness of this approach by (i) applying the adult framework on an independent more homogeneous glioma cohort and (ii) transferring the pediatric models to two different pediatric subtypes (ependymoma and medulloblastoma). In both cases, the data fusion model showed better performance for prognosis prediction than using each modality in isolation. This suggests that data fusion is a promising strategy to address the differences between diagnostic categories in one unified model.

[1] Gravendeel, L.A.M., et al., Intrinsic Gene Expression Profiles of Gliomas Are a Better Predictor of Survival than Histology. *Cancer Research*, 2009. 69(23): p. 9065-9072.

2. I had the impression that the clinical data was used in the fusion classifiers, in addition to the transcriptome and image data, but this data is not shown in the results.

In the previous submission we indeed mentioned the use of clinical data as a third modality, but as this third modality didn't add anything to the performance of the multimodal models (lines 420-422 in initial submission), we didn't include the results in the main manuscript and only focused on the data fusion of histopathology with expression data. Note that in the analyses of this revision, we decided to not include clinical factors, and solely investigate the data fusion of histopathology with expression data and the performance of each single modality versus the multimodal models.

3. Were the histopath only classifiers trained on the entire dataset, or the subset that also had transcriptome data available?

The "histopathology only" models were only trained on the subset of samples that also had expression data available. Likewise, the "expression only" models were only trained on samples that had histopathology images available. This to make a fair comparison between models and assessment of performance when joining the data for these samples.

4. When selecting images, how do you account for non-informative tiles – e.g. tiles that span blood vessels, tissue folding, crushing, areas of bleeding, or even tiles without tissue?

In our framework, each slide is first preprocessed to segment the target tissue from the background (OTSU image segmentation) [1]. After this tissue segmentation step, the non-overlapping patches with at least 20% tissue are extracted from the foreground using the OpenSlide library [2], and we randomly sample up to 100 of these patches per slide. Note that in this revised work we now used FFPE images instead of FF images (see below in comment number 7).

[1] Otsu, N., A Threshold Selection Method from Gray-Level Histograms. IEEE Transactions on Systems, Man, and Cybernetics, 1979. 9(1): p. 62-66. 10.1109/TSMC.1979.4310076

[2] Goode, A., et al., OpenSlide: A vendor-neutral software foundation for digital pathology. J Pathol Inform, 2013. 4: p. 27. 10.4103/2153-3539.119005

5. The authors report that there was a 50/50 split for the ependy/medullo datasets (line 178, methods). Did the authors train models based on these small datasets? If so, I couldn't see where they were reported. I suppose that these models didn't perform well but if mentioned in the methods, they should be shown.

The reviewer is right that a 50/50 split doesn't make a lot of sense for these small cohorts. In our new experiments, we did not split these cohorts for model training but performed transfer learning directly on the whole set of samples (see Methods section "Transfer Learning" Lines 407-419 and Results section "Transfer Learning Between Pediatric Brain Tumor Subtypes Lines 497-515). This is one of the main motivations for this manuscript, to show how multi-modal models can be transferred to cancer sites with less available samples (such as pediatric brain tumor subtypes).

6. For tumours with >1 slide available, does that potentially bias their contribution to the predictor, since more tiles/patches will be considered for these patients in comparison with patients with only 1 slide available?

This is a good question and we are happy to elaborate more on this. It is indeed true that there are more patches available for patients with more than 1 slides (for each slide 100 random patches are selected during training). However, while model training was patch-based (i.e. the model aims at predicting a survival score for each patch), during the evaluation phase the risk scores of all patches from the same patient are averaged to obtain one final patient's risk score eliminating any bias. In other words, the patient-level

strategy used to evaluate the model between training iterations, makes that the number of slides is not relevant.

7. For the flash-frozen tissue sections, in my experience, they provide less resolution and detail than FFPE-sections, and it might be more informative to analyse the FFPE samples. Do you think this could be problematic?

We fully agree and thank the reviewer for making this remark. Instead of Fresh Frozen (FF) samples, we retrained the histopathology model on images from Formalin-Fixed Paraffin-Embedded (FFPE) throughout the revised manuscript. As the reviewer correctly pointed out, compared to FF, FFPE better preserves cellular and architectural morphology. FF on the other hand, is more ideal for gene expression analysis (better RNA quality) than FFPE. Therefore, fusing FF with expression data might not add a lot of additional value, explaining why the multimodal models didn't show a significant improvement in performance. As such, the morphological features from FFPE data might indeed better complement corresponding expression data and can thus be more informative when fusing these two different modalities. As described in the result section (Lines 435-482) and summarized in Table 1 and Figure 5, data fusion of FFPE data with the expression data indeed lead to significant increased performance versus each single modality alone.

8. Transfer learning –
 - a. given that the other tumour cohorts are already small (47 ependy, 60 medullo), it is difficult to justify the 50:50 split into train/test prior to model validation and are arguably too small to assess prognostic capability. How were the train/test sets applied in this study? I didn't see that the ependy and medullo predictors were assessed?

As already noted above (reviewer's comment 5), it is entirely correct that a 50/50 split doesn't make a lot of sense for these small cohorts. In our new experiments, we did not split these cohorts for model training but performed transfer learning directly on the whole set of samples (see Methods section "Transfer Learning" Lines 407-419 and Results section "Transfer Learning Between Pediatric Brain Tumor Subtypes Lines 497-515).

- b. If the authors think that the application of the transfer learning to the ependy and medullo sets is justified, the Kaplan-Meier curves should be shown, in addition to the Cox model outputs so that the significance of the predictor can be assessed.

In this revised work, based on comments during review we have simplified the results and we only performed transfer learning from the pediatric glioma models

to the other two pediatric cohorts, i.e. ependymoma and medulloblastoma. Together with other updates in this revision (e.g. using FFPE, cross-validation approach and transfer of the model to the whole cohort instead of a subset), this led to a more robust performance. Important to note here is that the main purpose of our transfer learning experiments was to evaluate the performance on the entire new cohort, i.e. if the extracted features can be meaningful for this cohort. And thus, not the performance on each specific sample as the exact risk scores are only meaningful when the last layer model can be fine-tuned for on the new cohort (also see comment 8(c)). In addition, as the ependymoma and medulloblastoma cohorts only consist of high-grade samples especially the Brier score is of interest (absolute error measure), while the concordance index is less useful as a performance measure in a homogeneous cohort. Also, the Kaplan-Meier curves are less insightful when performing transfer learning as the determined cutoffs are cohort-specific and therefore only applicable for patients within the training cohort.

- c. While ependymoma is a glial tumour, it is difficult to see the rationale for the transfer learning applied to medulloblastoma, a neuronal tumour that has its own distinct clinico-molecular disease behaviour that is very different to gliomas and this decision should be justified in more detail.

It is indeed true that medulloblastoma stands out from the other brain tumors with its own characteristics and different origin. However, this is actually one of the goals of the work presented in this manuscript, to test whether multimodal models can be transferred to other diseases, even beyond brain tumors. Transfer learning is particular of interest diseases with a small sample size making it hard to develop a model from scratch. Transfer learning can counteract this limitation by re-using a model trained on a first task (with an abundance of available data) to a second related, but different task. This pre-trained model is then used as a starting point for the new task of interest. There are two main approaches: (i) direct transfer, and (ii) fine-tuning. In the latter approach, the weights and features of the base model are further refined on the new samples (this can be the whole model or only parts of it), while in the first approach the base model is directly “off-the-shelf” used on the new data. Here, we opted for a direct transfer given the limited number of samples in the ependymoma and medulloblastoma cohorts (and model fine-tuning needs splitting the data in train/test sets). The key reasoning behind direct transfer learning (but transfer learning in general), is that the knowledge and features learned on the first dataset might also be of interest for the second dataset. In our experiments, we were interested in investigating how the features learned on the pediatric glioma cohort, both for single and multimodal models, would perform when used on different brain tumors. Because of the fact that ependymoma is a glial tumor, one would indeed expect better performance for this cohort. Looking at the results in Table 3, it is indeed the case

that transferring the glioma model to ependymoma has slightly better scores over medulloblastoma. However, also on this distinct pediatric brain tumor, the transfer learning perform, especially for the early fusion models, postulating that in addition to glioma-specific features this framework also learns more general brain tumor features that could be relevant in other brain tumor types. The effectiveness of transfer learning demonstrated in our framework is potentially helpful for survival predictions in a more general context, where there is heterogeneity in the patient cohorts, as well as in the context of pediatric brain tumor subtypes with small cohort sizes where it is intractable to develop de novo models.

9. The first results deal with the creation of classifiers from the adult and paediatric glioma cohorts.
 - a. For adult glioma, remarkably the very simple age/sex classifier performs equivalently to the more complex predictors created – if an age/sex classifier performs comparably, what is the advantage of invoking these more complex predictors? However, for the paediatric glioma sets, the RNA-data alone and one of the fusion classifiers perform equivalently and are far superior to the clinical model.

It is indeed true that the model based on the age and sex features performed well on our adult glioma cohort in the first iteration of our work. However, as pointed out by other reviewers; this could be an artifact and we expect this is not the case for other (cancer) cohorts including brain tumors, as seen by the pediatric cohorts as well as in the transfer learning experiments. The same is true for the RNAseq of the pediatric cohort; while it performs well on the training cohorts it does not transfer well to other cohorts. Single data modalities are prone to overfit the training cohort. As also mentioned in the response to comment 1, brain tumors are very heterogeneous in both morphological and genomic features with variable prognosis. To tackle this variability, integration of multiple data sources is needed that can handle this heterogeneity. Importantly, combining more complex data modalities provides the opportunity to identify biological meaningful processes, as exemplified by the results of our pathway analysis (lines 518-535), and to detect new cancer driver genes. Also, while in this work we assessed the outputs generated by our models on cohort-level, these model outputs of combining multiple data sources can be potentially used as future biomarkers after determining the correct threshold with additional validation studies. In addition, our proposed multimodal framework using more complex data modalities is not brain cancer specific but can be used for other diseases.

- b. Why do the Kaplan-Meier curves in figures 4 and 5 not include the curves for the late fusion model?

We have reworked the figures of the manuscript and have now also included the Kaplan-Meier curves of the late fusion models for both the adult and pediatric cohorts (Figure 4).

10. For the paediatric glioma sets, the predictors pick out a group of patients with favourable risk and a 100% survival – it would be good to see at-risk tables to see the differences between the number of patients assigned to the favourable risk group in each panel, since the favourable-risk curves look similar.

In the right panel of Figure 4, we show the survival curves and the separation between the low- and high-risk groups of the pediatric glioma cohort. As the reviewer requested, below are the same curves for each model, but now with the risk tables indicated below each figure.

Figure: Kaplan-Meier curves of the pediatric glioma test set with risk tables. (A) Histopathology model. (B) Gene expression model. (C) Early or Feature fusion model. (D) Joint fusion model. (E) Late fusion model.

11. The transfer learning between the adult and paediatric glioma datasets should also be accompanied by Kaplan-Meier plots (at least in the supplementary data), along with the accompanying Cox models, to enable the reader to assess the models directly. The Brier scores and Concordance indices are noticeably lower, so perhaps the models do not transfer particularly well.

Also see the same response for comment 8(a). In this revised work, based on comments during review we have simplified the results and we only performed transfer learning from the pediatric glioma models to the other two pediatric cohorts, i.e. ependymoma and medulloblastoma. Together with other updates in this revision (e.g. using FFPE, cross-validation approach and transfer of the model to the whole cohort instead of a subset), this led to a more robust performance. Important to note here is that the main purpose of our transfer learning experiments was to evaluate the performance on the entire new cohort, i.e. if the extracted features can be meaningful for this cohort. And thus, not the performance on each specific sample as the exact risk scores are only meaningful when the last layer model can be fine-tuned for on the new cohort (also see comment 8(c)). In addition, as the ependymoma and medulloblastoma cohorts only consist of high-grade samples especially the Brier score is of interest (absolute error measure), while the concordance index is less useful as a performance measure in a homogeneous cohort. Also, the Kaplan-Meier curves are less insightful when performing transfer learning as the determined cutoffs are cohort-specific and therefore only applicable for patients within the training cohort.

12. In the latter part of the paper, the authors back-propagate the predicted risk to identify features important for outcome prediction. Could you provide details on how this was done in the methods, I didn't see where it was mentioned?

The specific methodology on how the back-propagation and pathway analysis was done was indeed poorly covered in the methods section. We have now added a dedicated paragraph "Pathway Analysis" (Lines 421-432) describing the steps to calculate the gene gradients and how the pathway analysis was performed.

13. The histology data shown in Figure 6 are interesting, but the indicated areas in figure 6a (top left of the right-hand panel), appear to be non-tumour (perhaps neuropil, though resolution is low). Is this problematic? Which tumour type is figure 6? Could the survival be ascribed to the upper panel (a) having larger nuclei compared to panel (b)? Large nuclei are typically associated with poorer-survival.

In our framework, we used ResNet-50 architecture to model the histopathology images. ResNet-50 is one of most popular deep learning networks and can be trained/finetuned for many computer vision tasks. Also for digital pathology, they have achieved accuracies

higher than classical ML models [1]. However, they remain “black-boxes” that do not directly reveal the features that they use to make predictions. Luckily there exist some methods that try to decompose the network into learned features. This is what we aimed to illustrate in this specific figure, by showing the regions or pixels that the model uses to make prediction from the FF slides. In such visualizations, it’s important to know that the features that ResNets use to make predictions are often hard to interpret and need to be distinguished from biological features. These features are not the same as the usual hand-crafted features, like shape of cells, but consist of abstract patterns and concepts from raw pixels [2]. Note that in this revised work, we used FFPE instead of FF as input of the histopathology models as FFPE better preserves cellular and architectural morphology. However, also accommodating comments raised by other reviewers about the biological significance (or lack thereof) of these visualizations, we decided to not include them in this revision.

[1] Noorbakhsh, J. et al. Deep learning-based cross-classifications reveal conserved spatial behaviors within tumor histological images. *Nat. Commun.* 11, 1–14 (2020)

[2] Foroughi pour, A., White, B.S., Park, J. et al. Deep learning features encode interpretable morphologies within histological images. *Sci Rep* 12, 9428 (2022). <https://doi.org/10.1038/s41598-022-13541-2>

14. In figure 7, the same approach is applied to the transcriptome data, and specific reactome pathways with non-random gradient scores are shown. Is this result significant? How do the pathways compare versus genes chosen at random? It would be good to see some significance scores if possible to assess whether these are meaningful. Have the most important genes previously been associated with survival in glioma and/or the other tumour types?

We have now added a more elaborate discussion on the found pathways/genes in the results section (Lines 518-543) and discussion (Lines 594-610) and their importance in cancer and/or glioma. In summary, for both the gene expression model and the joint model, known cancer pathways are found including cell cycle pathways linked to poor prognosis, and pro-apoptotic signaling related to good prognosis. For example, PTK6 directed cell cycle activity and NTRK2/3 activation through RAS, RAC1 and CDK5 pathways are identified as poor prognosis pathways consistent with previous reports [1,2,3]. Contribution of RUNX2 and RUNX3 expression is also observed. RUNX genes are frequently deregulated in various human cancers, indicating their prominent roles in cancer pathogenesis including glioma [4,5]. Interestingly, while signaling and regulation of RUNX transcription factors are amongst the top pathways of the joint fusion model, these observations are not prevalent in the RNA only model. This suggests that the joint fusion model benefits from the combination of the histopathology and expression data to uncover relevant cancer pathways

The performed pathway analysis differs from a “standard” pathway analysis in a couple of ways. While a standard pathway analysis is a kind of enrichment analysis of genes that are different between groups coupled to statistical testing, the foundation of our method are the pathway gradients related to the sample prediction. In other words, which pathways are important for a neural network when making a prediction for a particular data point. Fetching the gradients is a technique used to try to explain how each input feature affects the model output with the aim to increase transparency and interpretability of machine learning models. Pathway gradients were obtained by first performing a backpropagation of the prediction to get the gene gradients and next mapping the genes to the pathways of the Reactome pathway collection [6]. Importantly, gene gradients differ from gene expression values and represent the contribution in making the actual prediction. For each pathway, a gradient was obtained by averaging the gradients of its gene set. This was done for all samples. By visualizing the pathway gradients for all samples in a SHAP summary plot [7], it is possible to assess what pathways were the most important (or contributed the most) in making the predictions. By plotting the full distribution of a pathways’ importance over all observations we get a much better idea of its effect than could be captured by a single number (p-value). On that note, however, it’s not impossible to obtain single metrics for significance. In theory, P-values could be obtained in a non-parametric way by bootstrapping, i.e. retraining the model many times on resamples (bootstraps) of the training data. Each bootstrap model can then be used on the original data to calculate the pathway gradients and ultimately calculate a p-value per pathway. However, while not impossible, for complex models on a large dataset this approach is very time consuming and computational exhaustive.

[1] Pozo, K. and J.A. Bibb, The Emerging Role of Cdk5 in Cancer. *Trends Cancer*, 2016. 2(10): p. 606-618. [10.1016/j.trecan.2016.09.001](https://doi.org/10.1016/j.trecan.2016.09.001)

[2] Ito, K., et al., PTK6 regulates growth and survival of endocrine therapy-resistant ER+ breast cancer cells. *NPJ Breast Cancer*, 2017. 3: p. 45. [10.1038/s41523-017-0047-1](https://doi.org/10.1038/s41523-017-0047-1)

[3] Jin, W., Roles of TrkC Signaling in the Regulation of Tumorigenicity and Metastasis of Cancer. *Cancers (Basel)*, 2020. 12(1). [10.3390/cancers12010147](https://doi.org/10.3390/cancers12010147)

[4] Vladimirova, V., et al., Runx2 is expressed in human glioma cells and mediates the expression of galectin-3. *J Neurosci Res*, 2008. 86(11): p. 2450-61. [10.1002/jnr.21686](https://doi.org/10.1002/jnr.21686)

[5] Sun, J., et al., RUNX3 inhibits glioma survival and invasion via suppression of the beta-catenin/TCF-4 signaling pathway. *J Neurooncol*, 2018. 140(1): p. 15-26. [10.1007/s11060-018-2927-0](https://doi.org/10.1007/s11060-018-2927-0)

[6] Gillespie, M., et al., The reactome pathway knowledgebase 2022. *Nucleic Acids Res*, 2022. 50(D1): p. D687-D692. [10.1093/nar/gkab1028](https://doi.org/10.1093/nar/gkab1028)

[7] Lundberg, S.M. and S.-I. Lee, *A Unified Approach to Interpreting Model Predictions*, I. Guyon, et al., Editors. 2017.

15. The brief discussion recapitulates what was done, but I was confused with the discussion regarding the number of patches – in the methods, lines 374-7, it states that a number of different patches were used to train the model, but that

you settled on 100 patches. In the discussion, lines 569-572, the authors state that there is a limit of 2-3 patches per slide in practice. Could you clarify what you mean here? The discussion is too short, and should consider the study limitations in more depth, and also consider whether this approach could augment existing clinical markers.

The reviewer is indeed correct in that we settled on 100 patches per slide as training data for the model. Former lines 569-572 of the discussion are indeed confusing and we thank the reviewer for pointing this out as we could have done a better job in explaining this more. In general, the discussion was too short and didn't provide a thorough assessment of the performed analysis, their interpretation, implications as well as limitations and food for future work. We have rewritten the discussion including these aspects and hope it is more comprehensive. Regarding the limit of 2-3 patches per slide; this refers to the number of patches used to represent a slide during training. Recall that in our framework model training is patch-based, i.e. the model aims at predicting a survival score for each patch. This is done for all the 100 patches for each slide. However, during model evaluation (testing), the risk scores of all patches from the same patient are averaged to obtain one final patient's risk score (see Lines 255-270). As such, during training we do not leverage multiple patches to obtain the prediction for one WSI. One reason we only use 1 patch per slide during training is hardware limitations. If multiple patches per slide would be used and merged to obtain predictions during each training iteration/batch, the total number of slides per training batch would need to be reduced. However, the Cox loss requires many slides per training batch to learn the differences in survival times. Therefore, given the computational restraints, it was not possible to increase the number of used patches per slide during training. So in summary, given our computational resources, we could only use 1 patch per slide for each training batch. In situations with more computational power, this number could be increased and merged during training cycles but in practice a maximum of 2 to 3 patches per slide is reached for this patch-based approach.

Reviewer #5:

In their manuscript entitled “Multi-modal data fusion of adult and pediatric brain tumors with deep learning”, Qiu and colleagues developed a multimodal deep learning model and trained it on a combination of histopathological and RNA expression data to predict the prognosis of patients with pediatric and adult brain tumors. To this end they developed three different strategies for data fusion and training of the network. They looked at different pediatric as well as adult brain tumor types and used one data set for each of these tasks. As performance metrics, Harrel’s concordance index (C-Index) as well as the integrated Brier score (IBS) were used. Furthermore (as far as I understand), C-Index was used to divide the cohort into low- and high-risk patients for survival analyses. Lastly, image as well as RNA expression features were extracted and described. While the underlying idea -combining various data sources for more precise prognosis prediction- is innovative and promising, there are various points of concern:

1) Novelty

In my opinion, the study by Qiu et al. does neither substantially advance our knowledge about the use of multimodal deep learning models nor does it pose a significant improvement over previous studies. Similar approaches have been used for both brain cancer [1, 2] as well as other types of diseases [3-6]. What do the authors regard as substantially different to previous approaches?

It is indeed correct that multimodal data fusion has gained a lot of attention and that there have been other studies that proposed methods to fuse different data modalities, for disease prognosis, including cancer and glioma. Depending on the disease and the availability, multiple modalities are of interest. Importantly, each modality has its own characteristics and specific strategies are needed per modality, and for fusion of particular modalities. For example, in [3] they specifically focus on models for skin lesion pictures and clinical data, in [6] the focus is on fusing EHRs with clinical images, while our framework focuses on histopathology and genomic features.

The approaches in [1,2,4,5] also use histopathology and genomic features with [1,2] also for glioma prognosis, but we perceive that our proposed framework is more holistic and objective. For example, in [2], the authors describe a method that uses pathology images and genomic features. However, their work is mainly focused on survival prediction developing CNNs for the first modality only: pathology images. At a later stage they also present a framework that includes only a handful of preselected

genomic biomarkers. Here, there are a couple of important distinctions with our work: (i) they only use a handful of genomic biomarkers, (ii) these biomarkers originate from mutational data, and not expression data and (iii) they utilize a priori known biomarkers linked to cancer/glioma. Similarly, in [1,5], the authors start with pre-selected genomic biomarkers known as targets for brain and renal cancer, respectively. In contrast, our approach is a data driven framework using genome-wide (whole transcriptome) data and combining it with pathology images.

Thus, we argue that compared to these methods, our framework is novel as it is data-driven in that no prior selection or filtering based on known (cancer/glioma) biology. In fact, it's the opposite: we start from unbiased high-dimensional data as input for deep learning and the proposed modeling framework can discover potentially novel underlying biological processes (e.g. see Model Interpretation Lines 518-543). Thus, our approach has the potential to detect new cancer driver genes and also the methodology is not brain cancer specific but can be used for other diseases.

Next, we also perform a comprehensive analysis of different fusing strategies, ranging from early, joint to late fusion. We also introduce a new metric (composite score, see Lines 386-393) to evaluate the performance, and we validate on an external cohort and use transfer learning.

[1] Chen, R.J., et al., Pathomic Fusion: An Integrated Framework for Fusing Histopathology and Genomic Features for Cancer Diagnosis and Prognosis. *IEEE Trans Med Imaging*, 2020. PP.

[2] Mobadersany, P., et al., Predicting cancer outcomes from histology and genomics using convolutional networks. *Proc Natl Acad Sci U S A*, 2018. 115(13): p. E2970-E297

[3]. Liu, Y., et al., A deep learning system for differential diagnosis of skin diseases. *Nat Med*, 2020. 26(6): p. 900-908.

[4] Ning, Z., et al., Integrative analysis of cross-modal features for the prognosis prediction of clear cell renal cell carcinoma. *Bioinformatics*, 2020. 36(9): p. 2888-2895.

[5]. Schulz, S., et al., Multimodal Deep Learning for Prognosis Prediction in Renal Cancer. *Front Oncol*, 2021. 11: p. 788740.

[6] Huang, S.C., et al., Fusion of medical imaging and electronic health records using deep learning: a systematic review and implementation guidelines. *NPJ Digit Med*, 2020. 3: p. 136.

2) Methodology and Interpretation (Major)

- a. The biggest problem is that no independent external test set was used. It is completely unclear whether the fusion model is able to generalize and perform well on new data from different cohorts. We know that deep learning models are very good in “learning” (or better “memorizing”) cohort specific data structures, but then fail on similar data from external sources. This holds true for fusion models in particular as they tend to get rather “deep”. The necessity of using an independent external cohort is highlighted i. e. by Kleppe et al. and the authors should consider following the proposed PIECES recommendations and other AI guidelines [7]. Furthermore, a strong indication that the models are indeed overfitting the training data can be gathered from the fact that there is a dramatic drop in performance when models trained on pediatric tumors are used on adult tumors and vice versa.

7. Kleppe, A., et al., *Designing deep learning studies in cancer diagnostics. Nat Rev Cancer, 2021. 21(3): p. 199-211.*

We completely agree with the reviewer that validation on an external dataset is essential when performing DL experiments and evaluating performance. It is indeed known that deep networks are prone to overfit and often poorly generalize to other datasets [1]. As suggested by the reviewer, we additionally tested our adult glioma models on a third external validation set, CPTAC, an open-source dataset which holds FFPE and expression data of 97 glioblastoma (GBM) patients. Lines 484-496 and Table 2 show the results of the validation on this independent cohort.

[1] Bejani, M.M., Ghatee, M. A systematic review on overfitting control in shallow and deep neural networks. *Artif Intell Rev* 54, 6391–6438 (2021). <https://doi.org/10.1007/s10462-021-09975-1>

- b. It appears that the authors did not even use any type of k-fold cross validation either. They provide C-Index and IBS values for each modality together with some sort of confidence interval, but it is not clear where this confidence interval even comes from. It seems somewhat odd that the confidence interval is ± 0.02 for all experiments on adult glioma (Table 2). Where do these values originate from and how do the authors explain this? Furthermore,

there is no statement on whether the differences in performance (between the single modality models and the fusion models) are even statistically significant. To me the “improvement” by the combined approach seems only marginal and falls well within the margin of error (i. e. Table 1: RNA data alone: 0.78 ± 0.02 vs. Pathology + RNA: 0.79 ± 0.02). This holds true for each model presented throughout the entire manuscript. Just using a bold font does not make a model significantly better.

As correctly pointed out by the reviewer; multiple iterations of data splitting are needed to make a correct assessment of model performance and stability. In this revised work, we implemented a cross-validation strategy (10-fold for the adult cohort and 5-fold for the pediatric cohort -- because this pediatric cohort has less samples and the validation set would be too small for a good assessment of each fold). Models were evaluated on each validation set of the fold as well as on a separate test set that was not used for training (see methods section “Model Training” for more details (Lines 313-322)).

Next, we also updated the reporting on model performances. As described in the Methods section “Performance Metrics” (Lines 372-399), we defined a new scoring feature that combines the Integrated Brier Score (IBS) and Concordance-Index (CI) into one measure. This “Composite Score” (CS) thus represents an average of a relative measure (CI) and an absolute error metric (IBS) and allows straightforward evaluation and comparison of different models. Instead of just showing one score, model scores are now reported as mean and standard deviation (for CS, IBS, and CI scores) for the cross-validation experiments. In addition, we also performed a pairwise comparison (Wilcoxon rank sum) of the score distributions to evaluate the significance of differences in model performances.

Overall, we feel that the manuscript greatly benefits from the additional validation on the external cohort (see above), implementation of a cross validation strategy and the updated reporting. We hope that that this addresses the issues raised by the reviewer.

- c. C-Index as well IBS can only be used to evaluate the performance on an entire cohort of patients. For the purpose of clinical translation this is problematic as you cannot make predictions on an individual patient. This also limits the meaningfulness of the data presented with the Kaplan-Meier-Curves (Figures 4 and 5) as the presented cut-off would be highly specific only for patients within the cohort.

This is 100% correct and we completely agree with the reviewer that at this stage it's not yet possible to make individual predictions. However, the steps we did now and the visualizations we proposed are necessary valuable actions to proceed to such a level. The use of concordance index (CI), integrated brier scores (IBS) and now also our newly proposed metric "composite score" allow us to evaluate the effectiveness of the developed frameworks on cohort level. The fact that these measures give positive results gives us the confidence to use these frameworks to developed patient-specific biomarkers in future retrospective and prospective validation studies. While in this manuscript we assessed the outputs generated by our models on cohort-level, these model outputs can be potentially used as future biomarkers after determining the correct threshold with additional validation studies. The work presented here is the first step needed to go from bench to clinic.

- d. While the authors should be commended for including "explainability" experiments, the results presented in Figure 6 are not convincing. The image quality is poor, there are no scale bars, and neither an overview of the WSI nor a magnification of any cellular details are provided. Thus, the image features mentioned in the text cannot be recognized. The saliency maps do not point to a single feature (as would be expected) but are scattered throughout the entire image. The sarcomatoid morphology of the tumor shown in the figure cannot be recognized and is not highlighted by the heatmap. If this was the case, I would expect a much stronger signal colocalizing with the most pleomorphic cells and nuclei. If the other cells highlighted truly represent apoptotic cells, why exactly would the authors expect a poorer survival? Wouldn't an increase in apoptosis of the tumor be beneficial for the patient? A systematic analysis of the detected image features by an expert in(neuro)pathology is missing.

The reviewer is right that Figure 6 from the previous submitted work (saliency maps of histopathology model) was not that captivating, and the layout could have used some polishing. In our framework, we used ResNet-50 architecture to model the histopathology images. ResNet-50 is one of most popular deep learning networks and can be trained/finetuned for many computer vision tasks. Also for digital pathology, they have achieved accuracies higher than classical ML models [1]. However, they remain "black-boxes" that do not directly reveal the features that they use to make predictions. Luckily there exist some methods that try to decompose the network into learned features. This is what we aimed to illustrate in this specific figure, by showing the regions

or pixels that the model uses to make prediction from the FF slides. In such visualizations, it's important to know that the features that ResNets use to make predictions are often hard to interpret and need to be distinguished from biological features. These features are not the same as the usual hand-crafted features, like shape of cells, but consist of abstract patterns and concepts from raw pixels [2]. Note that in this revised work, we used FFPE instead of FF as input of the histopathology models as FFPE better preserves cellular and architectural morphology. However, also accommodating comments raised by other reviewers about the biological significance (or lack thereof) of these visualizations, we decided to not include them in this revision.

[1] Noorbakhsh, J. et al. Deep learning-based cross-classifications reveal conserved spatial behaviors within tumor histological images. *Nat. Commun.* 11, 1–14 (2020)

[2] Foroughi pour, A., White, B.S., Park, J. et al. Deep learning features encode interpretable morphologies within histological images. *Sci Rep* 12, 9428 (2022). <https://doi.org/10.1038/s41598-022-13541-2>

3) Methodology and Interpretation (Minor):

- a. It is unclear what type of ResNet was used and if the authors have tried other / newer architectures (DenseNets, EfficientNets, ViTs, etc.)? Furthermore, I don't really understand the reasoning behind not using any type of color normalization? Only using augmentation could be limiting performance.

We apologize if the methods section of the manuscript did not clearly define the used ResNet. We have now updated the text and made sure the specific type is included. Here, we used a ResNet-50 CNN architecture [1] for the histopathology model. Resnet-50 CNNs have already proven their value as a good model for clinical images, also in brain tumors [2,3].

However, as the reviewer correctly points out, the last years other and newer CNN architectures became available, such as DenseNet, EfficientNets amongst others [4]. These newer models are usually more complex and often come with additional computation time, cost, and potential memory limitations. ResNets on the other hand provide a good balance between computation needs and model performance. But, if hardware limitations are not an issue, it could be worthwhile to explore some newer CNN architectures, especially for unimodal model development. As multimodal fusion approaches already have additional computational needs, ResNet architectures provide a good compromise between too complex and too simple models. As we found this

feedback very useful, we have also included these thoughts to our discussion (Lines 626-635). When working with H&E slides some kind of normalization or additional augmentation is indeed essential for performance to account for variability and H&E stains. In our work, we opted for a stain augmentation without additional normalization, this for two reasons: (i) normalization adds significantly more computational needs, and (ii) in [5] the authors show that augmentation is the most important for CNNs and that additional normalization adds limited value on top, and We also added this reasoning to the manuscript in the methods section (Lines 210-214).

[1] He, K., et al. Deep Residual Learning for Image Recognition. 2015. arXiv:1512.03385.

[2] Sahaai, M.B., et al., ResNet-50 based deep neural network using transfer learning for brain tumor classification. AIP Conference Proceedings, 2022. 2463(1): p. 020014. 10.1063/5.0082328

[3] Chhabra, M. and R. Kumar, An Efficient ResNet-50 based Intelligent Deep Learning Model to Predict Pneumonia from Medical Images. 2022. p. 1714-1721.

[4] Alzubaidi, L., et al., Review of deep learning: concepts, CNN architectures, challenges, applications, future directions. J Big Data, 2021. 8(1): p. 53. 10.1186/s40537-021-00444-8

[5] Tellez, D., et al., Quantifying the effects of data augmentation and stain color normalization in convolutional neural networks for computational pathology. Med Image Anal, 2019. 58: p. 101544. 10.1016/j.media.2019.101544

- b. Have the authors tried including WSIs from FFPE samples (sometimes referred to as diagnostic slides) of the TCGA cohort? Tissue morphology can present totally different in frozen sections as opposed to FFPE which again can influence a model's performance.

We fully agree and thank the reviewer for pointing this out. In addition to Fresh Frozen (FF) samples, we retrained the histopathology model on images from Formalin-Fixed Paraffin-Embedded (FFPE). Tissue morphology between these two types is indeed different. While FFPE fixation can result in cross-linking, degradation, and fragmentation of DNA, it has several advantages over FF such as preservation of the cellular and architectural morphology, easy storage, and availability [1]. FF on the other hand, is more ideal for gene expression analysis (better RNA quality) than FFPE. Therefore, fusing FF with expression data might not add a lot of additional value, explaining why the multimodal models didn't show a significant improvement in performance. As such, because FFPE better preserves cellular and architectural morphology, FFPE data might indeed better complement corresponding expression data and can thus be

more informative when fusing these two different modalities. As described in the result section (Lines 435-482) and summarized in Table 1 and Figure 5, data fusion of FFPE data with the expression data indeed lead to significant increased performance versus each single modality alone.

[1] Gao, X.H., et al., Comparison of Fresh Frozen Tissue With Formalin-Fixed Paraffin-Embedded Tissue for Mutation Analysis Using a Multi-Gene Panel in Patients With Colorectal Cancer. *Front Oncol*, 2020. 10: p. 310. 10.3389/fonc.2020.00310

- c. This might just be me, but I find it a bit of a stretch to refer to “age” and “gender” as “clinical data” in this context. I would usually use the term “clinical data” for other parameters such as blood pressure or EEG measurements. It would be much more interesting if the authors actually included other types of data such as radiologic images or laboratory parameters.

We agree with this comment of the reviewer and agree that using the term “clinical data” for these two features is far-fetched and might even be misleading. In this revised work, we have now removed this from the manuscript solely focused on the investigation of fusing histopathology with expression data and the performance of these two single modalities versus the multimodal models. We also agree that inclusion of more advanced clinical features such as blood markers and other imaging types radiographic images is an interesting extension but challenging as these multimodal data sets are not often available.

- d. The link to the provided Github repo does not work and I could not access the underlying code base.

We indeed added the wrong link to the GitHub repository. We have now added a “Data Availability” section (Lines 680-690) including the correct location of the GitHub repo containing the source code of the models.

Reviewers' comments:

Reviewer #1 (Remarks to the Author):

The authors have addressed all of my comments. Thank you.

Reviewer #2 (Remarks to the Author):

Overall the authors did a good job of responding to the comments from the prior review, (for example, testing the model on an additional external validation cohort, CPTAC, strengthens the study significantly), but a few points still need to be addressed.

Some of my comments from the first review need to be further addressed:

My comment 1 from the prior review asked for more details about the histopathological features extracted by the model. The author's response included removing prior Figure 6, which showed some examples of the histopathological features that were extracted by the model. I understand that interpreting the biological significance of such features is complicated and might fall beyond the scope of this paper, but I think they should include some examples, so that interested readers can get a better idea of what the model is extracting.

Along the same lines, I do not understand the author's response, which showed up multiple times in their response letter "However, also accommodating comments raised by other reviewers about the biological significance (or lack thereof) of these visualizations, we decided to not include them in this revision." Which reviewers' comments are they referring to? It seems like they did a good job of responding to some of the concerns about the histopathological features by replacing the FF images with FFPE images, so why not include some examples?

My comment 3 from the prior review asked how LGG and HGG are parsed between the low-risk and high-risk groups generated by the models. This question was not adequately addressed in their response or in the revised manuscript. For example, they should provide a supplementary table showing how many LGGs and HGGs were in the low-risk and high-risk groups for each model. It would also be helpful if they provided similar information about how the different specific diagnostic categories of glioma (GBM, IDH-wildtype, Astrocytoma, IDH-mutant, Oligodendroglioma-IDH-Mutant) parsed between the high and low-risk groups. I discuss this point in more detail below.

The first paragraph of the introduction needs to be revised to better match the current WHO classification for CNS tumors (5th edition), which now incorporates genetic alterations in the classification of gliomas and other CNS tumors. For example, gliomas with IDH mutations are distinct from IDH-wildtype gliomas, and these molecular features are also major determinants of the WHO grading-which is now applied within tumor types, and is indicated by Arabic numerals, not Roman numerals (minor detail).

The more important point is that there is a well-established classification of glioma that divides these tumors into prognostically significant groups on the basis of specific genetic alterations and that the designation of "GBM" and "LGG" is now interpreted within the context of these genetic alterations. The predictive strength of the multi-modal machine learning models should be

compared to that of the current clinical classification systems. Reviewer 4 in the prior reviews also raised this point, and in my opinion the authors have not adequately addressed this important issue. Specifically, they have divided their adult glioma cohorts into LGG and GBM, without providing the status of the key diagnostic/prognostic molecule features that are currently used to classify these tumors. It is well established, for example, that gliomas with IDH1 mutations and 1p/19q co-deletions (now both diagnostic criteria for oligodendroglioma) have a significantly better prognosis and longer survival. Since oligodendroglioma and astrocytoma have morphologic, genomic and prognostic differences, it is very possible that they distribute differently in the low-risk vs high-risk adult gliomas. Similarly, IDH1-mutant and IDH-1 wildtype gliomas may also distribute differently across the low-risk and high-risk adult gliomas. I think it is important for them to directly address these points.

Similar issues pertain to the pediatric gliomas, which are also classified on the basis of specific genetic alterations, such as diffuse midline gliomas, H3K27-altered. The prognosis of these tumors is affected by their locations as well as their intrinsic behavior, and it would be informative to know if such tumors were included in their cohorts and if so, how did they distribute in the low-risk vs high-risk groups.

Reviewer #4 (Remarks to the Author):

The authors are to be commended for a very thorough response to the detailed comments made to all reviewers, which in my view mean that the paper should now be accepted, subject to a minor comment and with some additional proof reading needed for picking up typos (I noticed FPPE instead of FFPE on line 155, and I thought there may be a typo in table 1, single modality, RNA, training set) - there may well be more that I missed.

The use of an independent validation set is a strength of the revised manuscript.

I think a comment should be made in the discussion regarding a point I raised in review regarding the comparison between these models and existing stratification models based on clinical and/or pathological/molecular features. Are these patch-based deep learning fusion models superior to current clinical practice?

Reviewer 3 Brief Comments:

Reviewer 3 had strong concerns regarding whether or not the more complex multimodal models did indeed outperform current, clinically-based stratifications. In their response to this reviewer, the authors outline how they implemented cross-validation within model development and how their models were applied to an external cohort.

In the updated results table, the authors define a composite score which combines concordance and Brier indices and provide a mean and standard deviation for the tested cross-validations. I don't have access to them at the moment, but I do see that the CI and IBS scores are shown in Tables S1 and S2.

It is good to see that there is a wholly separate external validation set as requested. The authors say that the CI is less important because all this cohort is High grade – is there no heterogeneity within outcomes for this HGG cohort?

Minor comment: Figure 4 – can you equalise the time interval i.e. convert days to months for the paediatric cohort?

In my view, the authors have addressed the criticisms raised by this reviewer.

Reviewer #5 (Remarks to the Author):

The authors have substantially improved the manuscript and addressed most of my concerns. They should be commended for their additional efforts. However, I cannot completely judge the changes as they have not been highlighted in the manuscript. It should also be discussed, how this paper differs from another recent publication by Chen et al. (PMID 35944502), and if the new findings / changes to the manuscript are “novel” enough to warrant publication.

There are some minor points I have come across:

- In Figure 4 the “Time” axis seems to be clipped / cut. Please show all data points even after 110 (left) and 2500 (right). Also the unit is missing.
- In Figure 5, why have the authors used the Wilcoxon test? If these values are originating from one type of experiment and / or are presented together some type of ANOVA or Kruskal-Wallis-test should have been used. Have the authors used correction for multiple testing? If not, why not?
- In Figure 6 the names of the pathways are hard to read and should be presented in a more appealing way.

Response to Reviewers

We thank the reviewers for their constructive feedback of our manuscript. We have reworked some sections of the manuscript and elaborated or added new sections and/or figures for further validation. Overall, we believe that the story of the manuscript has now been significantly improved.

Reviewers' comments:

Reviewer #1 (Remarks to the Author):

The authors have addressed all of my comments. Thank you.

Reviewer #2 (Remarks to the Author):

Overall the authors did a good job of responding to the comments from the prior review, (for example, testing the model on an additional external validation cohort, CPTAC, strengthens the study significantly), but a few points still need to be addressed.

Some of my comments from the first review need to be further addressed:

1. My comment 1 from the prior review asked for more details about the histopathological features extracted by the model. The author's response included removing prior Figure 6, which showed some examples of the histopathological features that were extracted by the model. I understand that interpreting the biological significance of such features is complicated and might fall beyond the scope of this paper, but I think they should include some examples, so that interested readers can get a better idea of what the model is extracting.

Along the same lines, I do not understand the author's response, which showed up multiple times in their response letter "However, also accommodating comments raised by other reviewers about the biological significance (or lack thereof) of these visualizations, we decided to not include them in this revision." Which reviewers' comments are they referring to? It seems like they did a good job of responding to some of the concerns about the histopathological features by replacing the FF images with FFPE images, so why not include some examples?

We fully concur with the reviewer that adding an interpretation of the models and their selected features has a lot of added value and is something that needs to be done, or at least make an attempt, to unravel what is going on under the hood, especially in a clinical context. We did this for the genomic part of the models, but we should have also added this for the histopathological aspect of the models. In the first submission, we included an example, but in hindsight, we feel that what we showed was not ideal and we could have done a better and more thorough job there (we showed the middle grid of the tissue slice with the model overlaid instead of focusing on the actual input patches of the model and how the model uses those to determine a risk scores for each patch and ultimately a risk score per patient). As such, similar as for the genomic models, we implemented an additional interpretability analysis for the histopathology models. First, we performed a qualitative analysis by visualizing the model gradients overlaid on the original 224x224 patches and evaluating what tissue regions contributed the most to the model output. Figure 7 shows examples of a bad survival sample and a good survival sample with the saliency map (model heatmap) for both the joint fusion and histopathology only model. This is shown for a patch that the model gave a high-risk and a patch for which the model provided a low-risk. To add some more (biological) significance / meaning, in a second step we performed a cell segmentation with HoverNet. Using this, one can visualize if the model puts more emphasize on (specific) tumor cells versus lymphocytes versus or other cell types or a combination, and which ones. After the cell segmentation, we did an follow-up quantitative analysis where we looked at the cell type distributions between bad survival and good survival samples (with high and low risk scores, respectively, according to the histopathology models). Interestingly, we saw that the clearest distinguishing cell types between these two groups are tumor cells and lymphocytes with a higher fraction of tumor cells in high-risk samples and a higher fraction of lymphocytes in low-risk samples, which were both found significantly different by a two sample t-test.

3. My comment 3 from the prior review asked how LGG and HGG are parsed between the low-risk and high-risk groups generated by the models. This question was not adequately addressed in their response or in the revised manuscript. For example, they should provide a supplementary table showing how many LGGs and HGGs were in the low-risk and high-risk groups for each model. It would also be helpful if they provided similar information about how the different specific diagnostic categories of glioma (GBM, IDH-wildtype, Astrocytoma, IDH-mutant, Oligodendroglioma-IDH-Mutant) parsed between the high and low-risk groups. I discuss this point in more detail below.

We respond to this comment below under point 5.

4. The first paragraph of the introduction needs to be revised to better match the current WHO classification for CNS tumors (5th edition), which now incorporates genetic alterations in the classification of gliomas and other CNS tumors. For example, gliomas with IDH mutations are distinct from IDH-wildtype gliomas, and these

molecular features are also major determinants of the WHO grading-which is now applied within tumor types, and is indicated by Arabic numerals, not Roman numerals (minor detail).

We appreciate that the reviewer pointed this out as this indeed an important piece of information that is also key in the context of this paper! We have now updated this paragraph mentioning that the revised WHO guidelines suggest combining histopathological features with molecular features to classify diffuse glioma. We also made note of some recent reports suggesting to also consider DNA methylation as an additional data layer in this context. Lastly, we corrected the roman numerals and now use Arabic numerals to indicate tumor grading, as this is indeed the latest WHO nomenclature, and thank the reviewer for bringing this to our attention.

5. The more important point is that there is a well-established classification of glioma that divides these tumors into prognostically significant groups on the basis of specific genetic alterations and that the designation of “GBM” and “LGG” is now interpreted within the context of these genetic alterations. The predictive strength of the multi-modal machine learning models should be compared to that of the current clinical classification systems. Reviewer 4 in the prior reviews also raised this point, and in my opinion the authors have not adequately addressed this important issue. Specifically, they have divided their adult glioma cohorts into LGG and GBM, without providing the status of the key diagnostic/prognostic molecule features that are currently used to classify these tumors. It is well established, for example, that gliomas with IDH1 mutations and 1p/19q co-deletions (now both diagnostic criteria for oligodendroglioma) have a significantly better prognosis and longer survival. Since oligodendroglioma and astrocytoma have morphologic, genomic and prognostic differences, it is very possible that they distribute differently in the low-risk vs high-risk adult gliomas. Similarly, IDH1-mutant and IDH-1 wildtype gliomas may also distribute differently across the low-risk and high-risk adult gliomas. I think it is important for them to directly address these points.

Similar issues pertain to the pediatric gliomas, which are also classified on the basis of specific genetic alterations, such as diffuse midline gliomas, H3K27-altered. The prognosis of these tumors is affected by their locations as well as their intrinsic behavior, and it would be informative to know if such tumors were included in their cohorts and if so, how did they distribute in the low-risk vs high-risk groups.

While our results show that multimodal frameworks perform better than using one single modality, the actual value in a clinical setting can indeed only be assessed by comparing it against current clinical standards for stratification. Therefore, we retrieved additional genetic and

diagnostic information about our adult glioma samples and assessed how the LGG and HGG samples are distributed amongst the low- and high-risk classes for each of our models (Supplementary Table S3). Here, instead of LGG and HGG we made use of a more detailed diagnostic classification system, ie Glioblastoma, Astrocytoma anaplastic, Astrocytoma NOS, Oligodendroglioma anaplastic, Oligodendroglioma NOS and mixed glioma. In this context, we also took the genetic information (IDH1 mutation status and 1p/19q codeletion) and evaluated how these classes are distributed amongst the poor survival and good survival groups, for each model (bottom part of Supplementary Table S3). Interestingly, these results also show the value of a multimodal approach in this context. For example, while majority of the glioblastoma samples are classified as high-risk in the FFPE only model (92%), the RNA model only classified 76% in the high risk group. On the other hand, for astrocytoma anaplastic (generally a high-grade brain tumor), the FFPE only model marks 8% of the samples as high risk, while the RNA only model finds 56% of these samples high risk. When we look at the multimodal models however, these show for both cases a good grouping in the high risk group and indeed seems to take leverage of both the histopathological and genomic information. The same can be said for the groupings of the genetic subtypes, where for example the FFPE only model classifies only 71% of the IHDwt samples into high risk group, but the RNA only and multimodal models approximately group 85% of these samples in the high risk group. In case of the pediatric glioma; the somatic mutation data was unfortunately not publicly accessible.

Overall, we want to thank the reviewer for his thorough assessment of our manuscript. We think that all the above and previous suggestions have strengthened our manuscript and provided a stronger foundation for validating our models.

Reviewer #4 (Remarks to the Author):

The authors are to be commended for a very thorough response to the detailed comments made to all reviewers, which in my view mean that the paper should now be accepted, subject to a minor comment and with some additional proof reading needed for picking up typos (I noticed FPPE instead of FFPE on line 155, and I thought there may be a typo in table 1, single modality, RNA, training set) - there may well be more that I missed.

We have performed an additional proofreading and indeed caught the typos the reviewer noticed as well as one additional one. We are grateful to the reviewer for pointing these out to us.

The use of an independent validation set is a strength of the revised manuscript.

I think a comment should be made in the discussion regarding a point I raised in review regarding the comparison between these models and existing stratification models based on clinical and/or pathological/molecular features. Are these patch-based deep learning fusion models superior to current clinical practice?

As suggested by the reviewer, we have now added an additional paragraph about the need for comparison of our models and existing clinical models. While our results show that multimodal frameworks perform better than using one single modality, the actual value in a clinical setting can indeed only be assessed by comparing it against current clinical standards for stratification. In this context, we also performed an analysis where we assess how our models classify poor and good survival samples into different diagnostic and genetic subtypes for adult glioma (Supplementary Table S3)

Reviewer 3 Brief Comments:

Reviewer 3 had strong concerns regarding whether or not the more complex multimodal models did indeed outperform current, clinically-based stratifications. In their response to this reviewer, the authors outline how they implemented cross-validation within model development and how their models were applied to an external cohort.

In the updated results table, the authors define a composite score which combines concordance and Brier indices and provide a mean and standard deviation for the tested cross-validations. I don't have access to them at the moment, but I do see that the CI and IBS scores are shown in Tables S1 and S2.

It is good to see that there is a wholly separate external validation set as requested. The authors say that the CI is less important because all this cohort is High grade – is there no heterogeneity within outcomes for this HGG cohort?

Yes we agree with this, while there is heterogeneity in survival outcomes for this HGG as this is a more homogeneous group, the absolute difference in survival times between patients is smaller. As such, the Concordance Index is a less suitable metric to evaluate the performance of our models since this performance metric scores how well the risk score ranks the patients based on the predicted order of each pair of samples and is this a relative performance measure. We feel that for a more homogeneous cohort, an absolute error metric such as the Brier score is a better more honest value to evaluate model performance.

Minor comment: Figure 4 – can you equalise the time interval i.e. convert days to months for the paediatric cohort?

We have updated the figure showing the Kaplan Meier curves, and have put the time scale in months for both cohorts. As also pointed out by reviewer 5, this updated figure now also contains all data points instead of a fixed end at 110 months on the x-axis.

In my view, the authors have addressed the criticisms raised by this reviewer.

Reviewer #5 (Remarks to the Author):

The authors have substantially improved the manuscript and addressed most of my concerns. They should be commended for their additional efforts. However, I cannot completely judge the changes as they have not been highlighted in the manuscript. It should also be discussed, how this paper differs from another recent publication by Chen et al. (PMID 35944502), and if the new findings / changes to the manuscript are “novel” enough to warrant publication.

Firstly, we wanted to apologize that the changes to the previous submission were not highlighted. The reason we did not include a tracked changes version is because the majority of the manuscript was completely re-written as there were five reviewers and each had many suggestions. As such we did a major revision with new and additional analysis to improve both the models and the manuscript, and each sentence would have been highlighted.

Regarding the recent publication by Chen et al; it is indeed true that this paper also proposes methods to fuse different data modalities for survival prognosis. They decided to use a pan-cancer approach, while we focused on brain cancer in both an adult and pediatric cohort. While their approach also uses histopathology and genomic features, we believe our proposed framework is more objective and data-driven. Prior to model building, Chen et al filtered the genomic data with a priori known cancer gene sets as well as CNV and mutational frequency of genes. In comparison, our framework is novel as it is data-driven in that no prior selection or filtering based on known (cancer/glioma) biology and let our proposed modeling framework discover potentially novel underlying biological processes. As such, we argue that compared to this and other methodologies, our approach has the potential to detect new cancer driver genes and also the methodology is not brain cancer specific but can be used for other diseases. In addition, we also perform a comprehensive analysis of different fusing strategies, ranging from early, joint to late fusion.

There are some minor points I have come across:

1. In Figure 4 the “Time” axis seems to be clipped / cut. Please show all data points even after 110 (left) and 2500 (right). Also the unit is missing.

We have updated the figure showing the Kaplan Meier curves, and have put the time scale in months for both cohorts (pointed out by reviewer 4) and added the time unit (months). As correctly pointed out by the reviewer, this updated figure now also contains all data points instead of a fixed end on the x-axis.

2. In Figure 5, why have the authors used the Wilcoxon test? If these values are originating from one type of experiment and / or are presented together some type

of ANOVA or Kruskal-Wallis-test should have been used. Have the authors used correction for multiple testing? If not, why not?

We initially used a Wilcoxon test because we wanted to indicate a local 1:1 pair-wise significance between each of the models instead of multiple pairwise comparisons testing for global significance. However, to accommodate the reviewer's correct suggestion, we also added the results of a Kruskal-Wallis test and the post-hoc Dunn test (with benjamini-hochberg multiple correction) (Supplementary Table S4). Also this test shows that the composite score distributions significantly improved predictive performance of the multimodal models over the unimodal models in the test set, with the early fusion strategy the best performing model.

3. In Figure 6 the names of the pathways are hard to read and should be presented in a more appealing way.

We agree with the reviewer that the used font and color of pathway names didn't make it easy to read some of the names. We have updated the figure using a new font with darker text color and removed all underscores.